# Genetic profiling of protein burden and nuclear export overload

Reiko Kintaka[1], Koji Makanae[2], Shotaro Namba[3], Hisaaki Kato[4], Keiji Kito[5], Shinsuke Ohnuki[6], Yoshikazu Ohya[6], Brenda J Andrews[1], Charles Boone[1,7], Hisao Moriya[2,4]*

[1]Donnelly Center for Cellular and Biomolecular Research, Department of Medical Genetics, University of Toronto, Toronto, Canada; [2]Research Core for Interdisciplinary Sciences, Okayama University, Okayama, Japan; [3]Matching Program Course, Okayama University, Okayama, Japan; [4]Graduate School of Environmental and Life Science, Okayama University, Okayama, Japan; [5]Department of Life Sciences, School of Agriculture, Meiji University, Tokyo, Japan; [6]Graduate School of Frontier Sciences, University of Tokyo, Tokyo, Japan; [7]RIKEN Center for Sustainable Resource Science, Wako, Japan

**Abstract** Overproduction (op) of proteins triggers cellular defects. One of the consequences of overproduction is the protein burden/cost, which is produced by an overloading of the protein synthesis process. However, the physiology of cells under a protein burden is not well characterized. We performed genetic profiling of protein burden by systematic analysis of genetic interactions between GFP-op, surveying both deletion and temperature-sensitive mutants in budding yeast. We also performed genetic profiling in cells with overproduction of triple-GFP (tGFP), and the nuclear export signal-containing tGFP (NES-tGFP). The mutants specifically interacted with GFP-op were suggestive of unexpected connections between actin-related processes like polarization and the protein burden, which was supported by morphological analysis. The tGFP-op interactions suggested that this protein probe overloads the proteasome, whereas those that interacted with NES-tGFP involved genes encoding components of the nuclear export process, providing a resource for further analysis of the protein burden and nuclear export overload.

*For correspondence: hisaom@cc.okayama-u.ac.jp

Competing interests: The authors declare that no competing interests exist.

## Introduction

Extreme overproduction of a gratuitous protein that has no cellular function causes growth defects, which, at least in part, appears to be caused by overloading the cellular resources for protein synthesis (*Dong et al., 1995*; *Snoep et al., 1995*; *Stoebel et al., 2008*; *Makanae et al., 2013*; *Shah et al., 2013*; *Kafri et al., 2016*; *Moriya, 2015*; *Eguchi et al., 2018*; *Scott et al., 2010*). This phenomenon is called the protein burden/cost and has been extensively studied in the budding yeast *Saccharomyces cerevisiae,* a model eukaryotic cell. Limiting functions defining the protein burden are thought to be the translational process upon nitrogen limitation, and the transcriptional process upon phosphate limitation (*Kafri et al., 2016*). The protein burden itself initially appears to be a relatively simple phenomenon, but little is known about the physiological conditions and cellular responses triggered by the protein burden.

To trigger the protein burden, a protein must be produced at a level sufficient to overload protein production resources (*Moriya, 2015*; *Eguchi et al., 2018*). This can happen only if the protein is otherwise harmless. Fluorescent proteins, such as EGFP, Venus, and mCherry, do not have any physiological activity in yeast cells and thus are considered gratuitous proteins. Therefore, these fluorescent proteins are believed to be produced at the highest possible levels in yeast cells, and their

overproduction triggers a protein burden (*Makanae et al., 2013*; *Kafri et al., 2016*; *Eguchi et al., 2018*; *Farkas et al., 2018*). Modifications to EGFP, such as adding a degradation signal, misfolding mutations, or adding localization signals, reduces its expression limit, probably because these modifications overload limited resources for the degradation, folding, and localization processes, respectively (*Geiler-Samerotte et al., 2011*; *Makanae et al., 2013*; *Kintaka et al., 2016*; *Eguchi et al., 2018*).

A recent study isolated a group of deletion strains in which growth defects upon overproduction of yEVenus are exacerbated (*Farkas et al., 2018*). Through the analysis of these strains, and conditions exacerbating the protein burden, the authors concluded that Hsp70-associated chaperones contribute to the protein burden by minimizing the damaging impact of the overproduction of a gratuitous protein. Chaperone genes, however, constitute only a relatively small fraction of the deletion strains isolated in the study, and thus the protein burden may impact numerous other processes.

To understand the physiological conditions caused by protein burden, we conducted a systematic survey of mutants that exacerbate or alleviate the growth inhibition caused by GFP overproduction. We surveyed genetic interactions between mutant strains and high levels of GFP overproduction (GFP-op) to genetically profile cells exhibiting this phenomenon. Here, if a mutation exacerbates growth inhibition by GFP-op, or if GFP-op exacerbates growth inhibition by the mutation, the mutation has a negative genetic interaction with GFP-op. Also, if a mutation alleviates growth inhibition caused by GFP-op, the mutation has a positive genetic interaction with GFP-op. If GFP-op relaxes the growth inhibition caused by the mutation, it is also detected as a positive genetic interaction.

To isolate mutant sets showing positive and negative genetic interactions with the protein burden, we used a condition causing significant growth defects due to high GFP-op from the *TDH3* promoter (TDH3$_{pro}$) on a multi-copy plasmid. In addition to a deletion mutant collection of non-essential genes, we surveyed temperature-sensitive (TS) mutant collections of essential genes. We performed a strict statistical evaluation to isolate mutants showing robust genetic interactions with high confidence.

We also attempted to distinguish between the protein burden and other process overloads by surveying genetic interactions between those mutant strains and a triple-GFP (tGFP) with a nuclear export signal (NES-tGFP). NES-tGFP triggers growth defects at a lower expression level than unmodified tGFP (*Kintaka et al., 2016*). If the protein burden can only be triggered by a harmless protein like GFP, mutants harboring genetic interactions with tGFP-op should be different from those with NES-tGFP-op, and the comparison of those mutants will identify consequences specific to the protein burden. Moreover, mutants harboring negative genetic interactions should contain limiting factors of the nuclear export and essential factors affected by the overloading of nuclear export.

## Results

### Isolation of mutants that have genetic interactions with GFP-op

To isolate mutants genetically interacting with GFP-op, we performed a synthetic genetic array (SGA) analysis (*Baryshnikova et al., 2010*; *Figure 1A*). As a query strain, we overproduced GFP (yEGFP) (*Cormack et al., 1997*) under the control of TDH3$_{pro}$ on the multi-copy plasmid pTOW40836 (*Figure 1B*). This plasmid contains two selection markers (*URA3* and *leu2-89*), and the copy number can be controlled by the culture conditions. The copy numbers of this plasmid under –Ura and –Leu/Ura conditions are around 10 and 30 copies per cell, respectively (*Eguchi et al., 2018*). While a strain harboring this plasmid shows growth defects even under –Ura conditions (*Figure 1C*), the strain shows more growth defects under –Leu/Ura conditions (*Figure 1D*), presumably because the copy number increase leads to an increase in GFP production, and probably causes a stronger protein burden-associated growth defect (*Eguchi et al., 2018*). The background principles that determine plasmid copy number and growth rate (genetic tug-of-war) are explained in detail in *Figure 1—figure supplement 1*.

We examined an array of 4323 deletion mutants in nonessential genes (DMA) and an array of 1016 conditional temperature-sensitive mutants (TSA) (*Costanzo et al., 2016*). Details on the calculation of genetic interaction scores from colony size are shown in *Figure 1—figure supplement 2*. We assessed the growth (fitness) of wild-type and mutant strains by measuring the colony size on agar plates under vector control and GFP-op conditions, respectively. This colony size was then

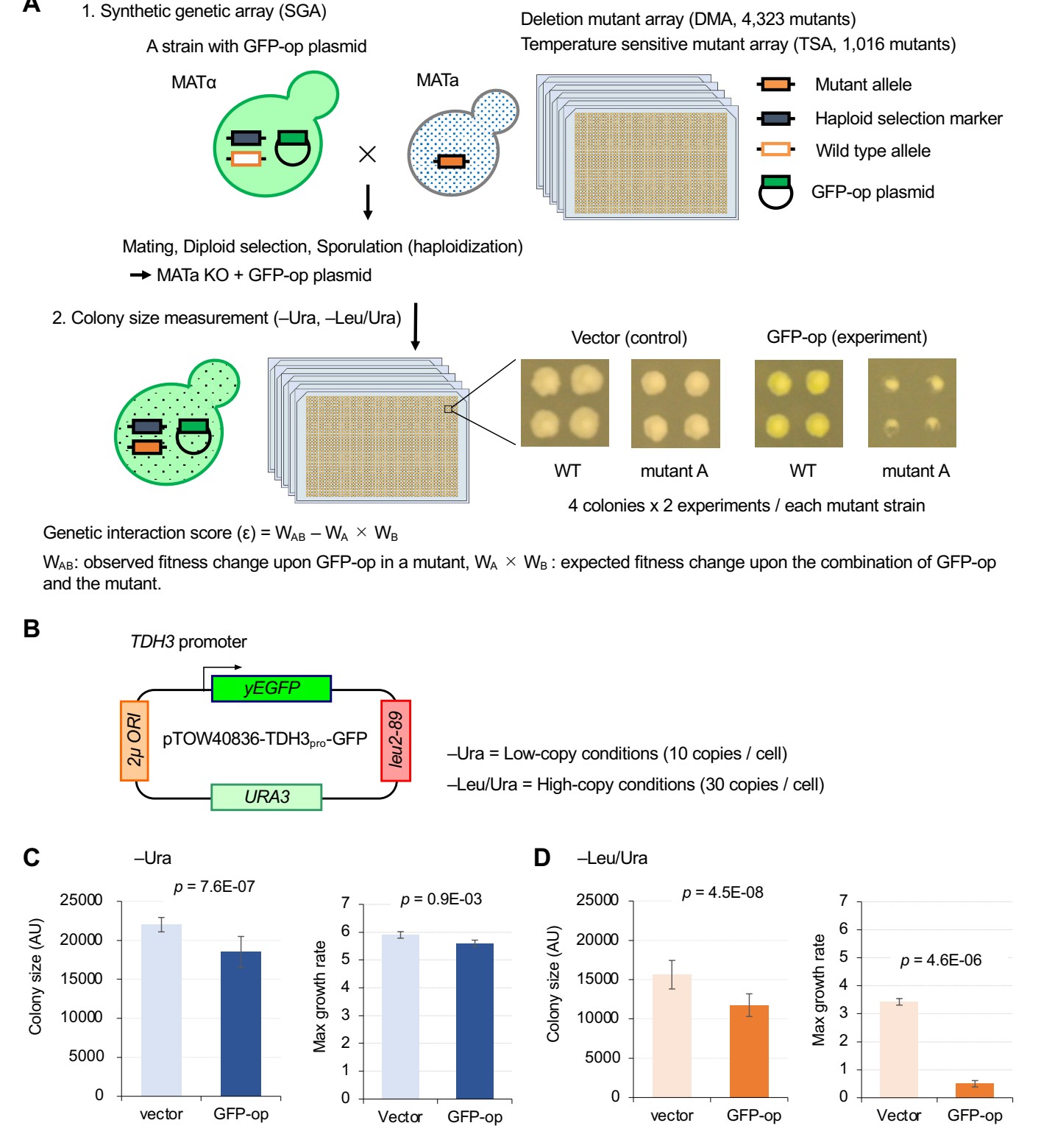

**Figure 1.** Experimental scheme of genetic interaction (GI) analysis. (**A**) Each mutant from a deletion mutant array (DMA) and a temperature-sensitive mutant array (TSA) was combined with GFP overproduction (GFP-op) using the synthetic genetic array (SGA) method (*Baryshnikova et al., 2010*). The colony size of each derivative strain grown on synthetic complete (SC)–Ura and SC–Leu/Ura plates was measured to calculate a genetic interaction (GI) score (ε). Four colonies were analyzed for each strain, and the entire experiment was duplicated. (**B**) The structure of the plasmid used to overexpress

*Figure 1 continued on next page*

*Figure 1 continued*

GFP. The plasmid copy number, and thus the expression level of GFP, can be changed by changing the growth conditions. (**C and D**) Effect of GFP production on growth under each condition. The size of colonies of each strain grown on agar medium was measured (n > 12). The Y7092 strain was used as the host. The maximum growth rate was measured in liquid culture (n > 4). The average, standard deviation (error bar), and *p*-value of Student's t-test are shown.

The online version of this article includes the following figure supplement(s) for figure 1:

**Figure supplement 1.** Background principles of genetic tug-of-war to determine plasmid copy number and growth rate.

**Figure supplement 2.** Measurement of fitness and calculation of genetic interaction score.

normalized by the overall colony size and the relative fitness of each strain was obtained as the normalized colony size. For each mutant strain, we calculated genetic interaction (GI) scores ($\varepsilon$) from the analysis of four colonies under both –Ura and –Leu/Ura conditions, in duplicate (***Figure 2—source data 1***). After thresholding by the variation in colony size (p<0.05), we compared GI scores between duplicates (***Figure 2A***, ***Figure 2—figure supplement 1***). The reproducibility of the DMA experiments was lower in –Ura conditions ($r = 0.17$), whereas it was higher in –Leu/Ura conditions ($r = 0.36$). The reproducibility of the TSA experiments was higher in both –Ura and –Leu/Ura conditions ($r = 0.42$ and $0.53$). Thus, the conditions which cause severe growth defects produce the most reproducible GI scores.

To more confidently identify mutants showing strong GIs, we set a threshold in each replicate ($\varepsilon > |0.08|$). Previous studies have reported that the use of these thresholds results in more reproducible genetic interactions (***Baryshnikova et al., 2010***). Using this threshold increased reproducibility, especially in the DMA experiments ($r = 0.35$ in –Ura, $r = 0.62$ in –Leu/Ura, ***Figure 2A***). We first selected mutants with $\varepsilon > |0.08|$ in each replicate and then calculated their average GI scores between the duplicates as summarized in ***Figure 2—figure supplement 2***. Because GI scores between –Ura (low-level GFP-op) and –Leu/Ura (high-level GFP-op) conditions were highly correlated ($r = 0.70$ and $0.58$, ***Figure 2B and C***), this procedure identified high-confidence mutants with GIs with GFP-op. We note that there is a higher correlation between conditions at –Ura and –Leu/Ura than between replicates in the DMA experiment (***Figure 2A*** and ***Figure 2B***). The cause of this is unclear, but it may indicate that averaging between replicates yields values closer to the true GI score.

Farkas et al. surveyed GIs between deletion mutants and the overproduction of yEVenus (***Farkas et al., 2018***). The GI scores obtained by our analysis did not show correlation with those from the Farkas study ($r = –0.01$ and $–0.07$, ***Figure 2—figure supplement 3A,B***). This may be because of the weak reproducibility observed in lower overproduction conditions (***Figure 2—figure supplement 1A***). Moreover this overlap analysis only involved nonessential genes and the Farkas study used a relatively weaker *HSC82* promoter (HSC82~pro~), in medium comparable to our –Ura condition, in which the GFP-op from HSC82~pro~ on pTOW40836 caused a very minor growth defect in –Ura conditions (***Figure 2—figure supplement 3E,F***). Indeed, our conditions produced more variance in the GI scores and thus identified more mutants showing stronger GIs (***Figure 2—figure supplement 3A,B***), and we found that negative GIs of 6 out of 7 deletion mutants from our screening were confirmed by independent growth measurements in the liquid medium, while all six mutants isolated by the previous study (***Farkas et al., 2018***) were not (***Figure 2—figure supplement 4***). Farkas et al. reported that growth inhibition (cost) due to overproduction of yEVenus was stronger as the concentration of amino acids in the medium decreased. As shown in ***Figure 2—figure supplement 3G***, overproduction of yEGFP also resulted in increased growth inhibition (cost) due to amino acid dilution, although the degree of cost differed between the two fluorescent proteins. We cannot dismiss the possibility that the difference between the analysis of Farkas et al. and ours is due to the difference in properties between our GFP (yEGFP) and yEVenus.

During the screening, we noticed that a group of temperature-sensitive mutants showed greater growth defects under –Leu/Ura conditions than under –Ura conditions in the vector control experiments (***Figure 2—figure supplement 5***). The gene ontology (GO) term 'DNA replication preinitiation complex [GO:0031261]' was significantly over-represented in the mutated genes (seven genes, p=1.47E–05). ***Figure 2—figure supplement 5A*** shows the normalized colony size differences of the 18 mutants analyzed in the TSA corresponding to the genes categorized in GO:0031261. 6 out of 18

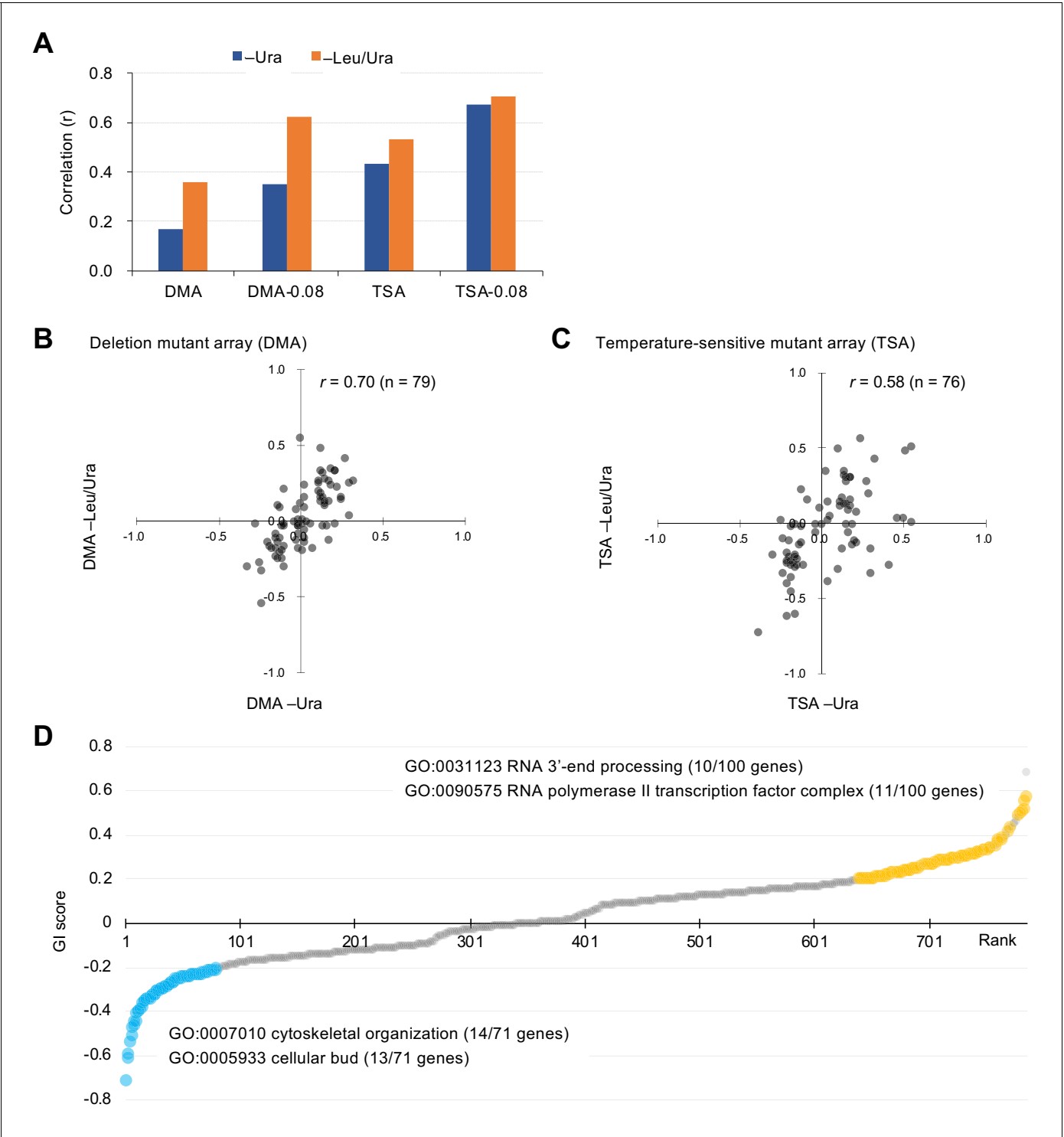

**Figure 2.** Characteristics of GI scores. (**A**) Pearson correlation coefficient (r) of GI scores from experimental duplicates. DMA and TSA: comparison of all GI scores of duplicates obtained by the GI analysis using DMA and TSA. DMA-0.08 and TSA-0.08: comparison of GI scores of duplicates with value > | 0.08| obtained by the GI analysis using DNA and TSA. *Figure 2—figure supplement 1* shows an independent comparison. (**B and C**) Comparison of average GI scores of DMA (**B**) and TSA (**C**) mutants both with GI scores in the duplicates > |0.08| under –Ura and –Leu/Ura conditions. (**D**) GI score (ε) of mutants isolated ordered by score ranking. Mutants with low (<0.2) and high (>0.2) scores are shown in light blue and orange, with enriched GOs in those mutants. The score in –LU is shown. The full list of enriched genes is in *Supplementary file 1*.

The online version of this article includes the following source data and figure supplement(s) for figure 2:

*Figure 2 continued on next page*

*Figure 2 continued*

**Source data 1.** Raw data of GFP-op SGA analysis; associated with *Figure 2A–C*; *Figure 2—figure supplement 1*; *Figure 2—figure supplement 5A*; *Figure 3*; *Figure 4—figure supplement 3D*; *Figure 5—figure supplement 1*.
**Source data 2.** Isolated GFP-op, tGFP-op, and NES-tGFP-op positive and _negative mutants by this study; associated with *Figure 2*, *4C*; *Figure 4D*; *Figure 5—figure supplement 2*; *Figure 5C*; *Figure 5D*; *Figure 5E*; *Figure 6A*.
**Figure supplement 1.** Comparison of GI scores from experimental duplicates.
**Figure supplement 2.** Scheme to isolate mutants showing GIs of high confidence with GFP-op.
**Figure supplement 3.** Comparison of GI analyses in this study and a previous study (*Farkas et al., 2018*).
**Figure supplement 4.** Verification of GIs with independent liquid growth measurement.
**Figure supplement 5.** Mutants of replication initiation complex specifically show growth defects in the high-copy conditions.

mutants showed more than 2U decrease in their colony sizes, whereas the average of all temperature-sensitive mutants showed 0.002U (Rep1) and 0.003U (Rep2). Colonies of representative mutants (*cdc47-ts*, *orc1-ph*, and *orc6-ph*) are shown in *Figure 2—figure supplement 5B*. The vector copy number is more than 100 copies per cell under –Leu/Ura conditions (*Makanae et al., 2013*; *Eguchi et al., 2018*). This high copy number probably produces limitations of the replication initiation complex by sequestering the complex to the replication origins of the plasmids (the explanation of the fitness reduction is shown in *Figure 2—figure supplement 5C*). Some negative factors on the plasmid, like $TDH3_{pro}$-*GFP*, restrict the plasmid copy number due to a genetic tug-of-war effect (*Figure 1—figure supplement 1C*; *Moriya et al., 2006*), and the plasmid thus may not trigger the limitation of the replication initiation complex. This situation may lead to a bias toward the isolation of mutants in the replication initiation complex with positive GIs with plasmids containing toxic elements, especially under –Leu/Ura conditions.

## Mutations aggravating or mitigating GFP-op triggered growth defects

To understand which processes are affected by GFP-op, we performed enrichment analysis targeted toward isolating mutants with stronger GIs ($\varepsilon > |0.2|$) under –Leu/Ura conditions, as the results obtained under these conditions were more reproducible (*Figure 2A*, *Figure 2—source data 1*). Therefore, we believed that stronger and more confidnet genetic interactions could be obtained with this threshold. We designated the negatively interacting genes and mutants 'GFP-op_negative' and the positively interacting genes and mutants 'GFP-op_positive'. The GFP-op_negaitive 71 genes (79 mutants) were significantly enriched in GO categories related to cytoskeletal organization and polarization (*Figure 2D*, *Supplementary file 1*). *Figure 3A* shows the GI scores under –Leu/Ura conditions of all 45 alleles of the GFP-op_negative genes categorized in GO as 'cellular bud [GO:0005933]'. Most of the mutants showed negative GIs, and 16 out of 45 showed average scores of less than –0.2.

One hundred GFP-op_positive genes (100 mutants) were enriched in genes involved in RNA 3'-end processing and the transcription factor complex (*Figure 2D*, *Supplementary file 1*). Among the factors in the RNA 3'-end processing, the subunits in the 'TRAMP complex [GO:0031499]' and 'nuclear exosome [GO:0000176]' were isolated as GFP-op_positive genes. *Figure 3B* shows the GI scores under –Leu/Ura conditions of the mutants of the TRAMP complex and the nuclear exosome subunits. Of 13 mutants, seven showed positive GIs with average scores greater than 0.2. Among the transcription factor complex, subunits of the 'mediator-RNA polymerase II preinitiation complex [GO:0090575]' were specifically isolated. *Figure 3C* shows the GI scores under –Leu/Ura conditions of the mutants of the mediator-RNA polymerase II preinitiation complex subunits. In total, 20 out of 38 mutants showed positive GIs with average scores greater than 0.2.

## Investigation of GFP expression levels of mutants

We next investigated GFP expression levels of each mutant overexpressing GFP. To obtain the GFP expression level of each mutant, we measured normalized GFP fluorescence (GFPunit) from the fluorescence intensity of each colony (*Figure 4A*). As summarized in *Figure 4B*, the GFPunit can be used to interpret the mechanisms underlying GFP-op_negative and GFP-op_positive mutations as follows: (1) if GFPunit is lower in a GFP_negative mutant, the mutant is considered to be more sensitive to GFP overproduction; (2) if GFPunit is higher in a GFP_negative mutant, the mutant triggers elevated GFP production, which potentially enhances protein burden; (3) if GFPunit is lower in a

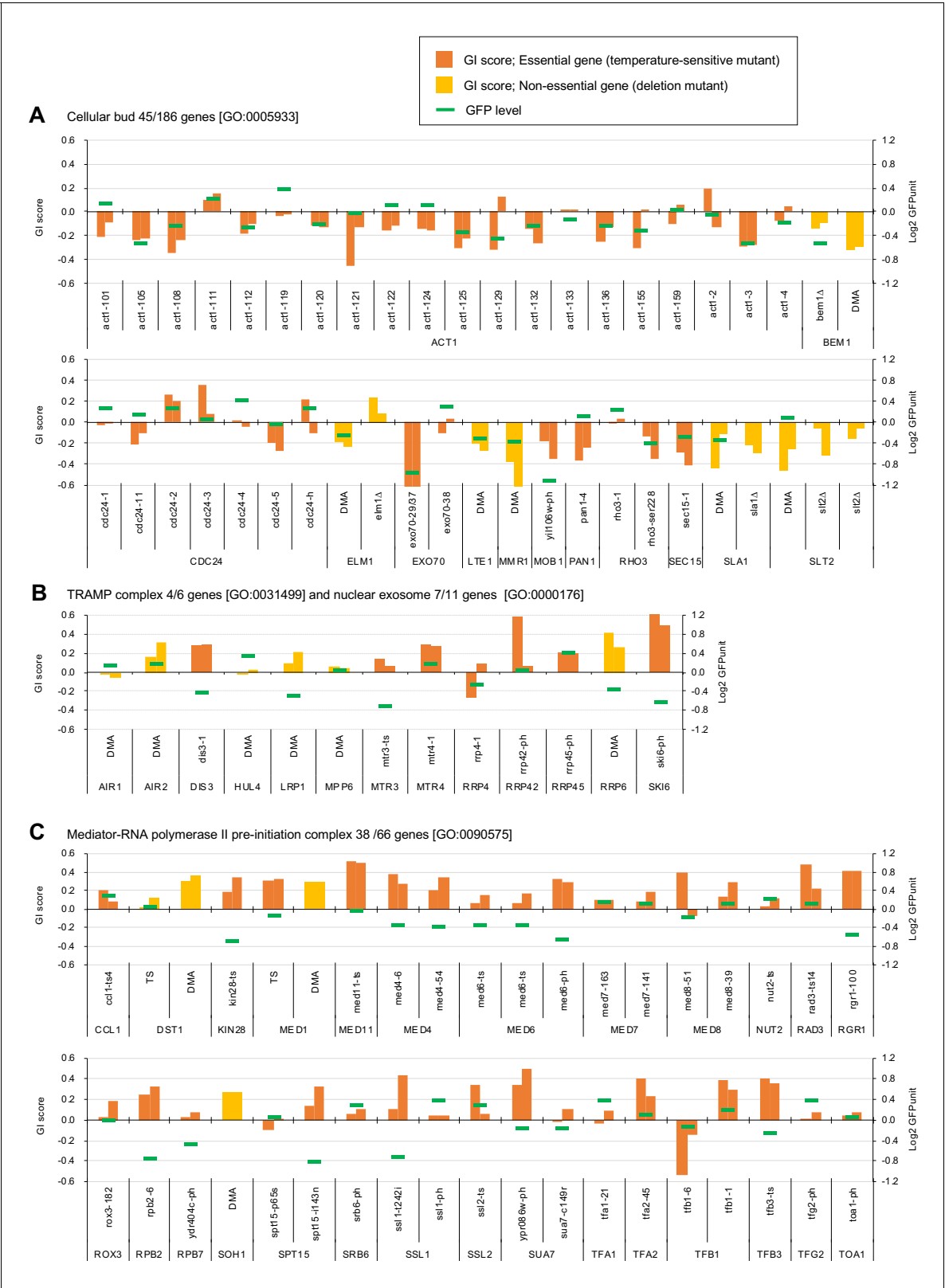

**Figure 3.** Independent GI scores (ε) of genes enriched in GO categories in GFP_negative and GFP_positive genes. (**A**) GI scores of mutants isolated as GFP_negative genes annotated with the GO term 'cellular bud [GO:0005933]'. (**B**) GI scores of mutants annotated with the GO terms 'TRAMP complex [GO:0031499]' and 'nuclear exosome [GO:0000176]'. (**C**) GI scores of mutants annotated with the GO term 'Mediator-RNA polymerase II preinitiation

*Figure 3 continued on next page*

*Figure 3 continued*

complex [GO:0090575]'. GI scores from experimental replicates under –Leu/Ura conditions are shown. Temperature-sensitive mutant of essential genes and deletion mutant of non-essential gene are shown in different colors. GFP levels for each strain are also shown.

GFP_positive mutant, the mutant triggers reduced GFP production, which potentially mitigates protein burden; and (4) if GFPunit is higher in a GFP_positive mutant, the mutant is considered to be resistant to GFP overproduction. The detailed background principle is explained in *Figure 4—figure supplement 1*.

Of the mutants, 1447 (29%) showed lower GFPunits and 3572 (71%) mutants show higher GFPunits than the average of all mutants (*Figure 4C,D*, *Figure 4—source data 1*). We designated these mutants GFP_H and GFP_L, respectively (*Figure 4C,D*). Mutants with enhanced growth defects upon GFP overproduction (GFP-op_negative mutants) were more likely to produce less GFP (GFP_L) (*Figure 4C*, p=4.7E-11, Student's t-test), indicating that the limit of GFP overproduction in these cells was lower than in other cells. Eleven out of 13 GFP-op_negative mutants categorized as 'cellular bud [GO:0005933]' were also GFP_L (*Figure 4—figure supplement 2A*, *Supplementary file 2A*) (the GFP levels for each strain in this category are shown in *Figure 3A*). These mutants seemed to be sensitive to the protein burden.

In contrast, only a slightly higher number of mutants in which the growth inhibition caused by GFP overproduction was alleviated (GFP-op_positive) had lowered GFP expression (GFP-L) (*Figure 4D*, p=0.013, Student's t-test). Trends in the distributions of mutants in 'TRAMP complex [GO:0031499]', 'nuclear exosome [GO:0000176]', and 'mediator-RNA polymerase II preinitiation complex [PMID27610567]' were not obvious (*Figure 4—figure supplement 2B*, *Supplementary file 2A;* the GFP levels for each strain in these categories are shown in *Figure 3*). However, GFP-op_positive and GFP_L mutants were significantly enriched in 'RNA polymerase II transcriptional factor complex [GO:0090575]', suggesting that these mutants may simply cause the reduction of GFP production, but not decrease the sensitivity to the protein burden.

The GFP-op_positive and GFP_H strains include strains resistant to GFP overproduction (i.e. protein burden). We recently reported one such strain, a deletion of the dubious gene *YJL175W*, which produces a partial deletion of *SWI3* (*Saeki et al., 2020*). We next searched for mutations where the mutation creates a growth advantage and where this advantage is further enhanced by GFP overproduction. We identified 14 mutants among GFP-op_positive and GFP-H mutants whose fitness in the vector control was higher than that of other strains and whose fitness may be further enhanced by GFP overproduction (*Figure 4—figure supplement 3*). The 14 mutants were significantly enriched in genes of the DASH complex [GO:0042729] (*Supplementary file 3*), including *ask1-2*, *dad2-9*, and *spc34-5* out of the seven DASH complex mutants analyzed (*Figure 4D*, *Figure 4—figure supplement 3C,D*). The DASH complex binds to microtubules and is involved in the distribution of chromosomes (*Jenni and Harrison, 2018*). At present, the molecular mechanism by which GFP overproduction is permissible in these mutants cannot be readily deduced and more detailed analysis is required.

## Overproduction of tGFP and NES-tGFP results in GIs with distinct sets of genes

We next analyzed mutants genetically interacting with a GFP containing a nuclear export signal (NES). Instead of GFP, we used tGFP made from three linked GFPs (*Figure 5A*) for the following reasons. In a previous study, we found that the addition of NES to monomeric GFP with a molecular weight smaller than the exclusion limit of the nuclear pore does not localize outside the nucleus, and when the molecular weight is increased by linking three GFPs together, the extranuclear localization of tGFP is clearly established (*Kintaka et al., 2016*). In addition, probably because NES-GFP undergoes repeated free transport into the nucleus and transport out of the nucleus by means of transport machinery, overproduction of NES-GFP shows a very strong growth inhibition. And we have found that this growth inhibition is mitigated to some extent by using tGFP (*Kintaka et al., 2016*). If the growth inhibition is too strong, we cannot generate overexpressing mutants to detect genetic interactions. We used NES from PKI, and used the *PYK1* promoter, because the *TDH3* promoter is too strong and causes severe growth inhibition (data not shown). We also used PYK1$_{pro}$-tGFP as a

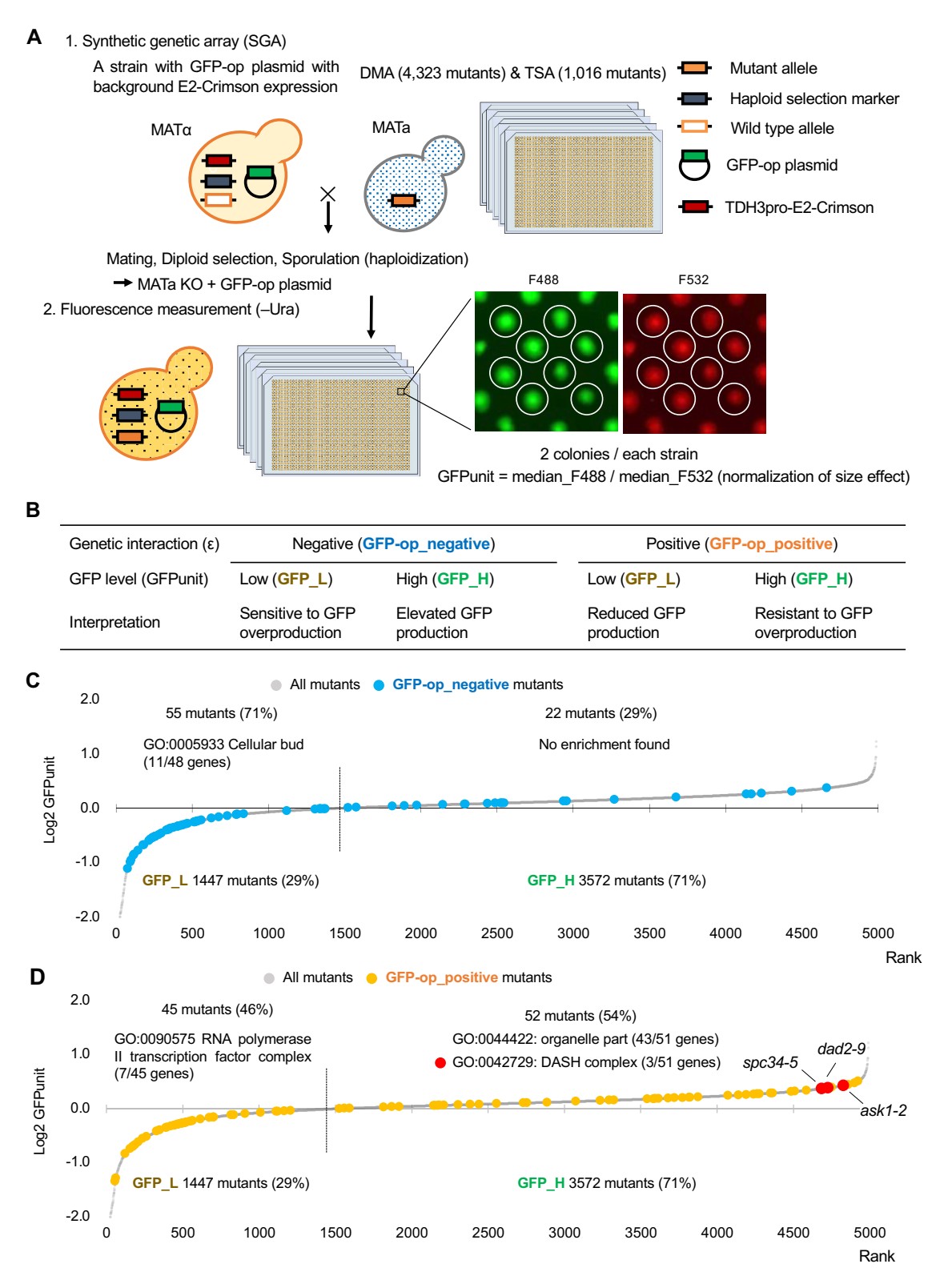

**Figure 4.** Experimental scheme of GFP expression measurements of mutants. (**A**) Each mutant from a deletion mutant array (DMA) and a temperature-sensitive mutant array (TSA) was combined with GFP overproduction (GFP-op) with background E2-Crimson expression, using a synthetic genetic array (SGA) method. The median GFP fluorescence (F488) and E2-Crimson fluorescence (F532) of each colony were measured, and the GFP expression level (GFPunit) of each mutant was calculated by dividing F488 by F532 to normalize colony size. (**B**) Further classification and interpretation of mutant strains

*Figure 4 continued on next page*

*Figure 4 continued*

exhibiting positive/negative genetic interactions using GFP levels. A more detailed explanation is shown in *Figure 4—figure supplement 1*. (C) GFPunits of GFP-op_negative mutants. Mutants with lower and higher GFPunits than the average are designated as GFP_L and GFP_H mutants, respectively. Representative GO terms enriched in GFP_L mutants in GFP-op_negative mutants are shown. (D) GFPunits of GFP-op_positive mutants. Representative GO terms enriched in GFP_L mutants and GFP_H mutants in GFP-op_positive mutants are shown. The full list of enriched genes is in *Supplementary file 2A*.

The online version of this article includes the following source data and figure supplement(s) for figure 4:

**Source data 1.** Raw data of GFP expression analysis under GFP-op, tGFP-op, and NES-tGFP-op; associated with *Figure 4—figure supplement 2A*; *Figure 4—figure supplement 2B*; *Figure 4C*; *Figure 4D*; *Figure 5—figure supplement 2*; *Figure 5E*; *Figure 6A*.

**Figure supplement 1.** Isolation of mutants using GFP expression levels and a detailed explanation of the background principles.

**Figure supplement 2.** Distribution of GFPunits of mutants in specific GOs among mutants with GI with GFP-op.

**Figure supplement 3.** Mutants of the DASH complex obtained as GFPop_positive increase the expression of GFP.

control for NES-tGFP (*Figure 5A*). Using the same procedures as in the analysis of GFP described above except upper and lower threshold of ε 0.16 and –0.12 as these thresholds have been used to obtain confident genetic interactions in previous studies (*Costanzo et al., 2010*; *Costanzo et al., 2016*), we isolated total 714 mutants (695 genes) harboring GIs with either GFP-op, tGFP-op or NES-tGFP-op under –Leu/Ura conditions (the raw data sets are in *Figure 5—figure supplement 1— source datas 1* and *2*, the isolated mutants are in *Figure 2—source data 2*). To extract genes that had specific GIs with each condition, we performed clustering analysis using them, which were isolated in at least one of GFP-op, tGFP-op, and NES-tGFP experiments (*Figure 5C*, *Figure 5—source data 1*).

*Figure 5D* shows the representative GO term or publication for each cluster (the whole data is shown in *Supplementary file 4*). Mutants negatively interacting only with NES-tGFP-op (Cluster 3) contained mutants of genes playing a central role in the nuclear protein export (Crm1, Gsp1, Rna1, and Yrb1). GI scores of these mutants were significantly lower in the NES-tGFP-op experiment than in the other two experiments (*Figure 5—figure supplement 1A*), suggesting that NES-tGFP-op specifically causes growth defects through overloading these limited factors.

To our surprise, only 12% (81/688) of mutants showed shared GIs between GFP and tGFP. Mutants negatively interacting with tGFP-op and NES-tGFP-op but not GFP-op (Cluster 4) were strongly enriched in annotations of 'cytosolic proteasome complex [GO:0031597]' (*Figure 5D*). GI scores of mutants in 'proteasome complex [GO:0000502]' were significantly lower in the tGFP-op and NES-tGFP-op experiments than in the GFP-op experiment (*Figure 5—figure supplement 1B*). These results suggest that GFP and tGFP have different characteristics, and tGFP-op triggers proteasome stress.

Mutants interacting only with GFP-op (Cluster 6 and Cluster 11) were enriched in genes annotated to 'cellular bud neck [GO: 0005935]' and 'transcription by RNA polymerase II [GO:0006366]', and their GI scores were significantly lower and higher in tGFP-op experiments than in the other two experiments (*Figure 5—figure supplement 1C,D*). This observation suggests that these two processes are specifically interacting with the protein burden and can be only triggered by proteins with very high expression.

We also measured tGFP and NES-tGFP expression levels in tGFP-op and NES-tGFP-op experiments using the same method as shown in *Figure 4*, and the results are shown in *Figure 5—figure supplement 2*. Among the mutants that had a negative genetic interaction with NES-tGFP-op, the mutants with lower GFP expression (NES-tGFP_L) were enriched for genes involved in nuclear protein transport (*Figure 5E*). Furthermore, all mutants in Cluster 3 in *Figure 5D* (*crm1-1*, *gsp1-162*, *rna1-1*, *rna1-s116f*, and *yrb1-52*) were included in this group (*Figure 5E*, *Supplementary file 2B*). As explained in *Figure 4—figure supplement 1B*, this result implies that these mutants are sensitive to high expression of NES-tGFP. The identification of nuclear protein transport mutants as specifically sensitive to overproduction of NES-tGFP can be considered as a proof of concept for this study, and the results strongly support that the analysis of GFP-op and tGFP-op also allowed us to obtain mutants specifically associated with the physiological state of the cells triggered by the overproduction of them.

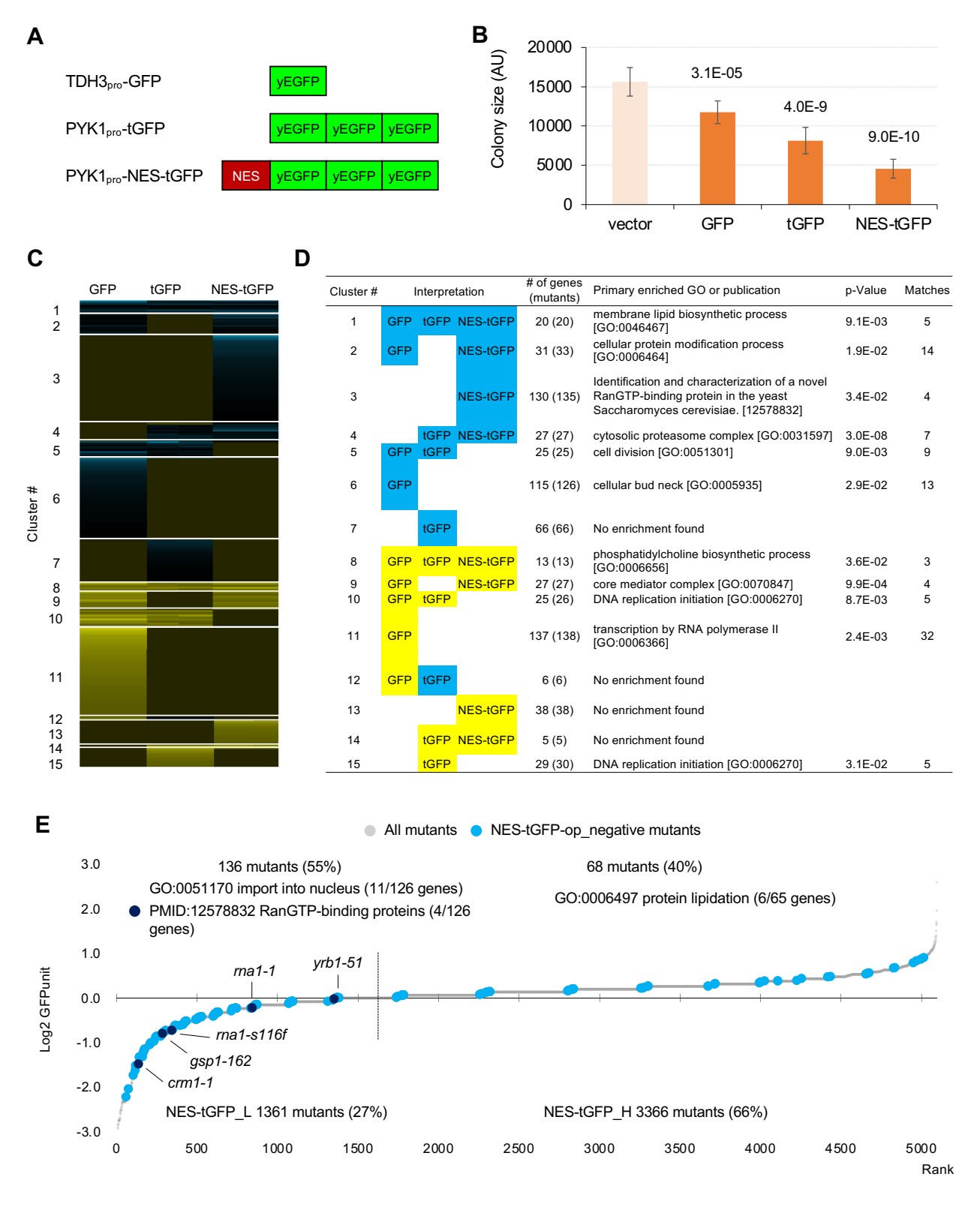

**Figure 5.** GFP-op harbor GIs with distinct sets of genes from those with tGFP-op and NES-tGFP-op. (**A**) Structures and promoters used to overexpress GFP, tGFP, and NES-tGFP. Nucleotide sequences of the three GFPs in tGFP (and NES-tGFP) are different, other to avoid recombination. (**B**) Colony size of query strains with the vector control and overproduction plasmids. The size of colonies of each strain grown on –Leu/Ura agar medium was measured (n > 6). The Y7092 strain was used as the host. The average, standard deviation (error bar), and p-value of Student's t-test are shown. (**C** and

*Figure 5 continued on next page*

Figure 5 continued

D) Clustering analysis of the mutants having GIs with GFP-op, tGFP-op, and NES-tGFP-op (C), and its characterization (D). Total 714 mutants (695 genes) with upper and lower threshold of ε 0.16 and –0.12 harboring GIs with either GFP-op, tGFP-op or NES-tGFP-op under –Leu/Ura conditions were used (*Figure 5—source data 1*). (E) GFPunits of NES-tGFP-op_negative mutants. Mutants with lower and higher GFPunits than the average are designated as NES-tGFP_L and NES-tGFP_H mutants, respectively. Representative GO terms enriched in GFP_L mutants in GFP-op_negative mutants are shown. Nuclear transport mutants obtained in Cluster 3 in D are also shown on the graph. The full list of enriched genes is in *Supplementary file 2B*.

The online version of this article includes the following source data and figure supplement(s) for figure 5:

**Source data 1.** Isolated mutants with GFP, tGFP, and NES-tGFP SGA analysis, and the result of clustering analysis; associated with *Figure 5C and D*.
**Figure supplement 1.** Distributions of GI scores of mutants in specific publications and GO.
**Figure supplement 1—source data 1.** Raw data of tGFP-op SGA analysis; associated with *Figure 5—figure supplement 1A–D*.
**Figure supplement 1—source data 2.** Raw data of NES-tGFP-op SGA analysis; associated with *Figure 5—figure supplement 1A–D*.
**Figure supplement 2.** GFPunits of mutants overproducing tGFP and NES-tGFP.

## Overproduction of triple GFP causes the formation of ubiquitinated intracellular condensates, which in turn may overload the proteasome

As noted above, we used tGFP as a control for NES-tGFP. We initially expected that tGFP was simply a protein consisting of three molecules of GFP linked together and having the same properties as monomeric GFP, and that their overproduction would show a similar genetic interaction profile. Because the amount of GFP expressed from one copy of tGFP is three times greater than that of GFP, we also predicted that the expression limit of tGFP would simply be reduced to one-third of the number of molecules as GFP. However, as mentioned above, the mutants that show a genetic interaction between the two were very different, suggesting that overproduction of tGFP causes a completely different effect than overproduction of GFP. In particular, as described above, only overproduction of tGFP (and NES-tGFP) showed a negative genetic interaction with the proteasome mutants (*Figure 5D*, Cluster 4).

Therefore, we next analyzed the properties of tGFP in more detail. *Figure 6A* shows the expression levels of tGFP in mutant strains that show negative genetic interactions with tGFP. Proteasome mutants were identified in both low (tGFP_L) and high (tGFP_L)-tGFP expression mutants. This suggests that overexpressed tGFP is actively degraded by the proteasome in the wild-type strains, whereas overaccumulation of undegraded tGFP occurs in proteasome mutations or that undegraded tGFP is cytotoxic in mutations of the proteasome.

If tGFP is actively degraded, then the amount of tGFP in the cell would be much lower than that of GFP. Indeed, when we investigated the abundance of GFP and tGFP by Western blotting, the amount of tGFP (and NES-tGFP) was about 4% of GFP (*Figure 6B*). As mentioned above, if GFP and tGFP have the same properties and generate protein burden in the same way, they should express about the same amount of protein units and about one-third of the number of molecules as GFP. Note that under these conditions (–Ura), where the protein levels were measured, cells harboring GFP, tGFP, and NES-tGFP plasmids showed a significant delay in the growth rate compared to the vector control (*Figure 6—figure supplement 1A*). From this, it can be said that the expression levels of these proteins have a negative effect on cell proliferation. Thus, tGFP (and NES-tGFP) was found to have different properties than GFP, as overproduction of only a few percent of GFP may trigger growth inhibition.

Next, we analyzed the behavior of overproduced tGFP in yeast cells. First, the subcellular localization of GFP, tGFP, and NES-tGFP was observed by fluorescence microscopy. We found that giant condensate were present only in the cells overproducing tGFP and NES-tGFP (*Figure 6C*). To further investigate the nature of this structure, cell fractionation was performed; the cell lysate was separated into soluble supernatant and insoluble precipitate. Fluorescence microscopy showed that the aggregates seen in tGFP-overexpressing cells were enriched in the precipitation (*Figure 6—figure supplement 1B*). Proteins in each fraction was then separated by SDS-PAGE and detected for GFP by Western blotting (*Figure 6D*). About 1% of GFP was found in the precipitation fraction, whereas 14% of tGFP was found in the precipitation fraction. Western blotting with anti-ubiquitin antibodies detected ubiquitinated high-molecular-weight proteins in the precipitates of tGFP-op (*Figure 6—figure supplement 1C*). We next identified the proteins in the precipitation by liquid

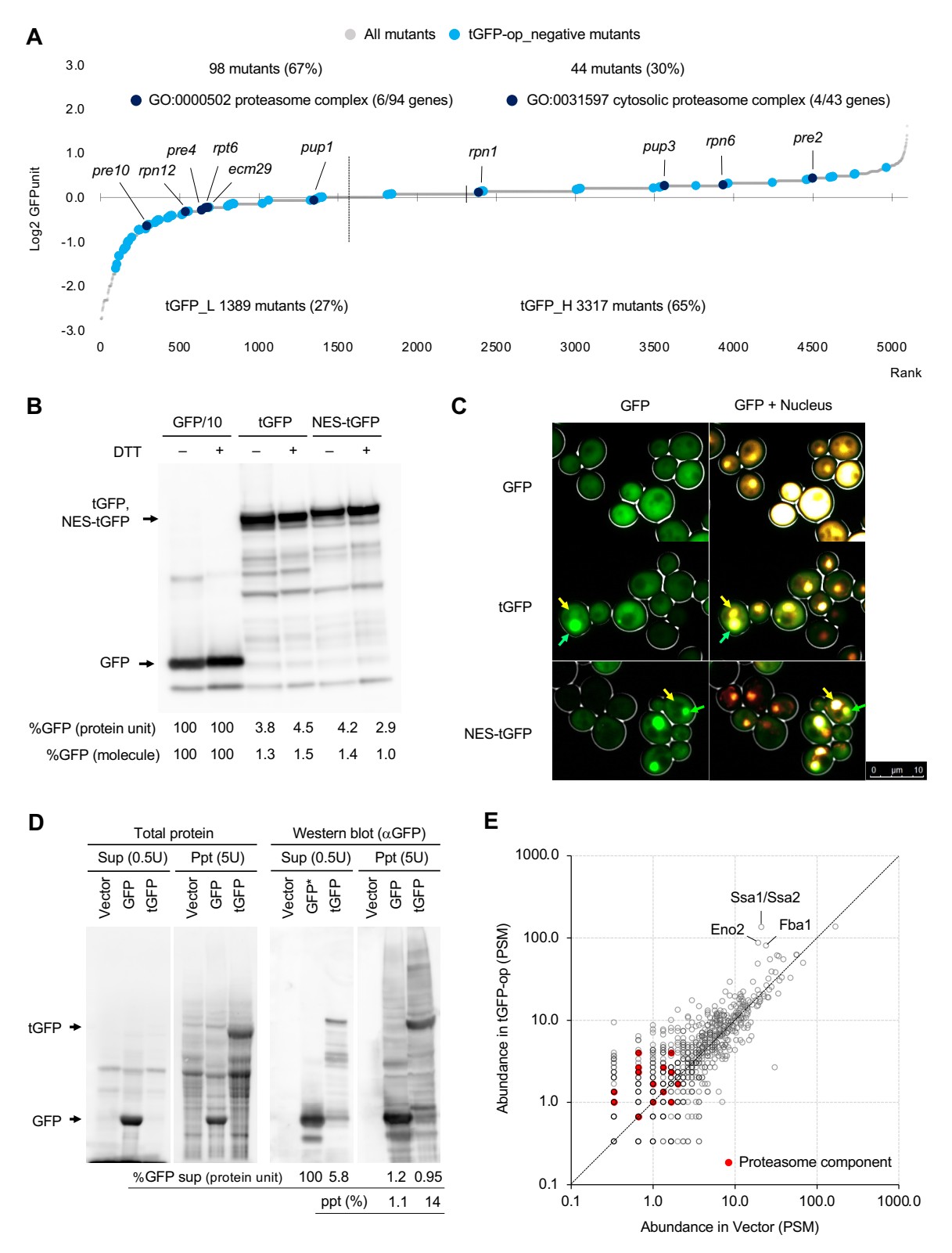

**Figure 6.** Overexpression of triple GFP causes the formation of intracellular condensates, which in turn may overload the proteasome. (**A**) GFPunits of tGFP-op_negative mutants. Mutants with lower and higher GFPunits than the average are designated as tGFP_L and tGFP_H mutants, respectively. Representative GO terms enriched in GFP_L mutants in GFP-op_negative mutants are shown. Proteasome mutants (excluding allele names) are also shown on the graph. The full list of enriched genes is in ***Supplementary file 2C***. (**B**) Quantification of expression limits of GFP, tGFP, and NES-tGFP.
*Figure 6 continued on next page*

*Figure 6 continued*

Western blot analysis of total protein from GFP-op (1/10 diluted), tGFP-op, and NES-tGFP cells cultured in SC–Ura medium. Relative GFP levels (protein units) were calculated by measuring the intensities of bands corresponding to the molecular weight of each protein (arrowheads). Note that molar concentration GFP should be divided by three in the case of tGFP and NES-tGFP because they have three times more epitopes for the antibody than GFP. (C) Microscope images of cells overexpressing GFP, tGFP, and NES-tGFP. The nucleus was observed using Hoechst 33342 staining. Representative cells with intracellular condensates are indicated by green arrowheads (condensates with GFP fluorescence) and yellow arrowheads (nucleus). (D) Analysis of GFP and tGFP aggregation. Extracts of yeast grown in SC–Leu/Ura medium were centrifuged, fractionated into supernatant (Sup) and precipitate (Ppt), and separated by SDS-PAGE. Electrophoretic images of all proteins and western blots with anti-GFP antibodies are shown. The molecular weights corresponding to GFP and tGFP are indicated by arrows. The amounts of GFP and tGFP detected in the supernatant and precipitation are shown with GFP in the supernatant as 100. The percentages of precipitation to the total are also shown. (E) Detection of proteins in precipitation fractions. Proteins in the precipitated fractions of the vector control and tGFP-op cells were detected by LC-MS/MS and their amounts were compared by peptide-spectrum match (PSM). Three proteins that were particularly abundant in the precipitation fractions of tGFP-op cells were indicated on the plot. The components of the proteasome (Ppn, Prt, and Pre proteins) are shown by red circles. The average values of three LC-MS/MS measurements are shown. The raw data are shown in *Figure 6—source data 1*.

The online version of this article includes the following source data and figure supplement(s) for figure 6:

**Source data 1.** Raw data of the LC-MS/MS analysis of proteins in the precipitation fractions of the vector control and tGFP-op cells; associated with *Figure 6E*.

**Figure supplement 1.** Growth inhibition by tGFP-op, and biochemical analysis of overexpressed tGFP.

chromatography-tandem mass spectrometry (LC-MS/MS). More than 1000 proteins were detected in the precipitation of the vector control and tGFP-op (*Figure 6E*). Among them, Hsp70 (Ssa1/Ssa2) and the glycolytic enzymes Fba1 and Eno2 were particularly abundant in the precipitates of tGFP-op. Components of the proteasome were also identified, albeit in trace amounts, and tended to be more abundant in the precipitation of tGFP-op than in the vector control. Ubiquitin was not detecte in this experiment.

## GFP-op affects actin distribution

The above results indicate that GFP-op, that is the protein burden, could affect actin functions. We thus performed a morphological analysis of cells under GFP-op with a high-throughput image-processing system (CalMorph) (*Ohtani et al., 2004*). We used non-fluorescent GFP mutant (GFPy66g) for this analysis because strong GFP fluorescence affects the observation of the cell shape with FITC-ConA. We also analyzed the cells overexpressing Gpm1 and a catalytically negative Gpm1 mutant (Gpm1-m) whose overproduction is considered to cause the protein burden (*Eguchi et al., 2018*). Cells were cultured under SC–Ura conditions. Among obtained 501 morphological parameters, only four parameters showed significant differences over the vector control, and three of them (A120_A1B, ACV7-1_A, and A122_A1B) were actin-related parameters (*Figure 7A–D*). *Figure 7E* shows the interpretation of the morphology of GFP-op cells. The cells contained increased actin patch regions, supporting the idea that the protein burden interacts with actin function.

## Discussion

In this study, we genetically profiled the consequences of protein overproduction using GFP as a model gratuitous protein and NES-tGFP as a transported model protein. We confirmed our prediction that the overproduction of NES-containing protein (NES-tGFP) overloads the amount of limiting nuclear-export factors (*Kintaka et al., 2016*). Overproduction of NES-tGFP had strong negative GIs with mutants in the major nuclear export factors (Crm1, Gsp1, Rna1, and Yrb1; *Figure 5D* and *Figure 5—figure supplement 1A*). tGFP-op (and NES-tGFP-op) had negative GIs with mutants in proteasome components but GFP-op did not (*Figure 5D* and *Figure 5—figure supplement 1B*). tGFP and NES-tGFP form aggregates, but not GFP (*Figure 6C*). This difference may be due to the higher molecular weight of tGFP compared to GFP and the presence of a repeating structure within the molecule. A larger molecule may increase the likelihood of misfolding during translation, or the presence of a repeating structure may increase intermolecular interactions and trigger the creation of large aggregates. Based on our results, we hypothesized a model explaining the negative genetic interaction between tGFP-op and the proteasome mutants (*Figure 7—figure supplement 1*). tGFP has a high probability of misfolding during translation, and when misfolded, it is ubiquitinated and

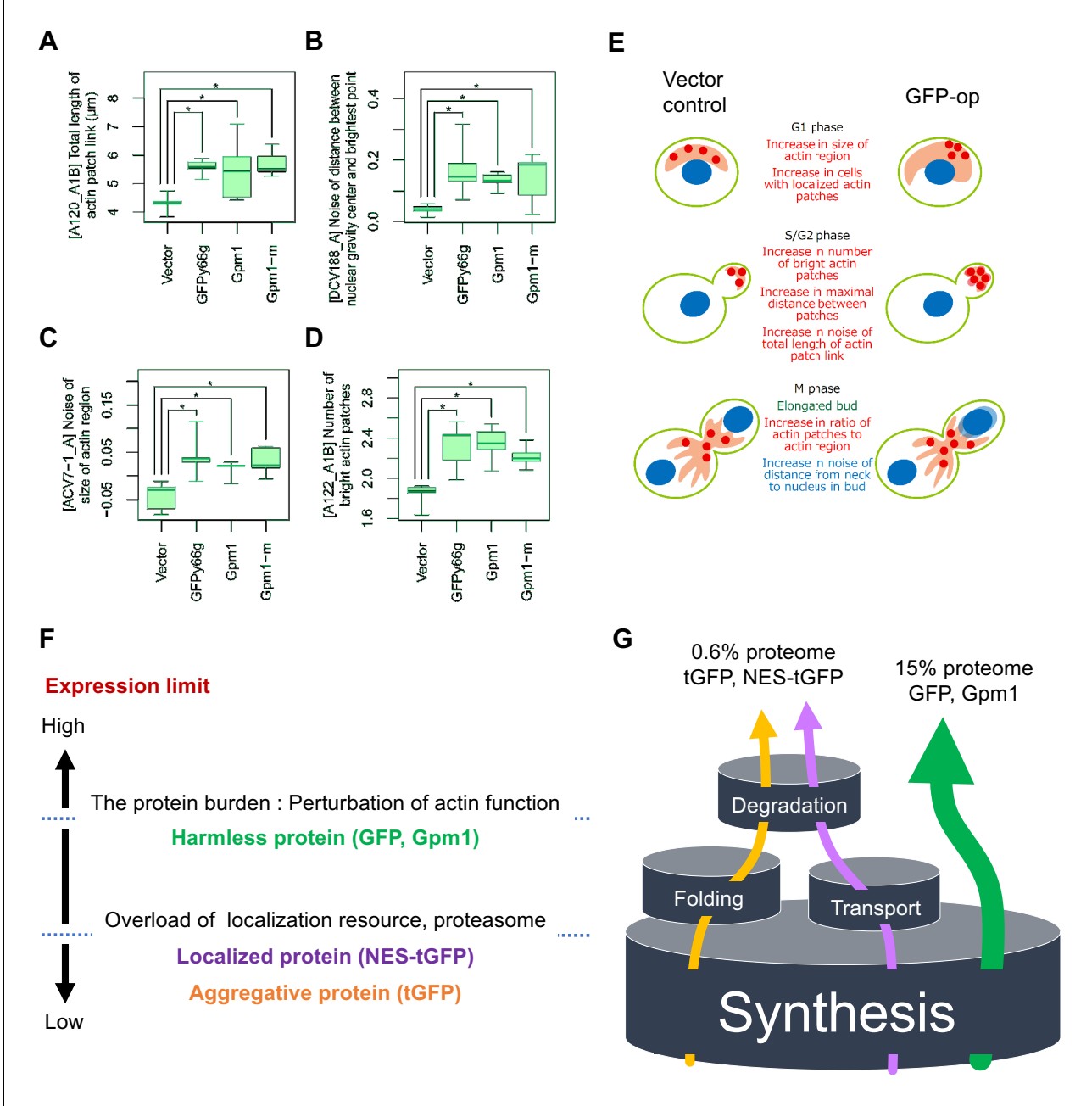

**Figure 7.** Morphological analysis of the cells overexpressing gratuitous proteins, and models explaining the consequence of overexpression. (A–D) Morphological parameters significantly different all in the cells overexpressing GFPy66g, Gpm1, and Gpm1-m cells over the cells with the vector control. *: FDR = 0.01 by Wald test. To overexpress GFPy66g, Gpm1, and Gpm1-m, pTOW40836-TDH3$_{pro}$-GFPy66g, pTOW40836-TDH3$_{pro}$-Gpm1, and pTOW40836-TDH3$_{pro}$-Gpm1-m were used. (E) Interpretation of the morphology of GFP-op cells according to the morphological parameters significantly different from the vector control. (F) Dissection of the consequence of protein overproduction by the expression limits. Only otherwise harmless protein could cause the protein burden, which is associated with the perturbation of actin function. (G) A 'barrel model' to explain the relationship between the capacity of intracellular processes and the limits of protein synthesis. An explanation of this model is described in Discussion. The online version of this article includes the following source data and figure supplement(s) for figure 7:

**Source data 1.** Whole dataset of morphological phenotyping of overexpressing strains; associated with *Figure 7A–D*.
**Figure supplement 1.** A hypothetical model of the effects caused by tGFP-op.

degraded by the proteasome (1). In addition, tGFP also forms large intracellular aggregates that sequester proteasomes (2), ubiquitin (3), and chaperones (4). Among these, 1 will overload the

proteasome's capacity (*Kintaka et al., 2016*). If 2 happens, it will lower the amount of proteasomes in the cell. If 3 happens, there would be a negative effect on proteostasis due to the depletion of ubiquitin (*Higgins et al., 2020*). The occurrence of 4 would have a similarly negative effect on proteostasis. In addition, since Ssa1/Ssa2 is required for proteasome assembly (*Hammack et al., 2017*), a depletion of these proteins would lower the amount of proteasomes. Based on the results of the proteome analysis, we believe that 4 is particularly likely, given that large amounts of Ssa1/Ssa2 were detected in the tGFP-op precipitates.

A comparison of mutants interacting with overproduction of three model proteins led to the isolation of mutants which specifically interact with GFP-op (*Figure 5*). The three model proteins caused growth defects with different expression levels (*Figure 6B*). The GFP level is considerably higher than the levels of tGFP and NES-tGFP, and its expression is the highest of all proteins in yeast (*Eguchi et al., 2018*), suggesting that overproduction of GFP causes growth defects because of the protein burden. As the protein burden should be triggered by the overproduction of otherwise non-harmful proteins like GFP (*Moriya, 2015*), these mutants should either exacerbate or mitigate the protein burden. The protein burden is considered to be growth defects occurring as a result of the overloading of protein synthesis processes (*Kafri et al., 2016*). In contrast to the expectation that mutants in those processes exacerbate the protein burden, the mutants isolated did not show any GO term enrichment in those processes but showed enrichment in actin-related processes like 'cytoskeletal organization' or 'cellular bud' (*Figure 2D*). Morphological analysis of cells also supported that GFP-op affects normal actin functions (*Figure 7A–E*). This relationship might be a result of the long-known connection between actin and translational machinery *Kim and Coulombe, 2010*; the protein burden-triggered growth defects might involve the perturbation of the actin cytoskeleton via translational factors like eEF1A, which can bundle actin fibers (*Munshi et al., 2001*). Mutations that mitigate the protein burden indeed enriched genes involved in protein synthesis, especially the transcriptional processes 'RNA 3′-end processing' and 'RNA polymerase II transcription factor complex' (*Figure 2D*). Because GFP expression levels in those mutants were lower than average (*Figure 4D*), those mutants might simply reduce the transcription of the GFP transcript itself.

It is thought that only harmless proteins can be produced up to 'the ultimate expression level' to cause the protein burden because harmful proteins should cause cellular defects at lower expression levels (*Moriya, 2015*). Those defects should be triggered by overloading more limited cellular resources, such as those used for folding and transport, accelerated non-specific interactions, or untimely activation of pathways (*Moriya, 2015*). Our study here supported this idea through the following observations: (1) tGFP (and NES-tGFP) consists of aggregates in the cell and thus could cause proteostasis stress (*Figure 6C,D*); (2) NES-tGFP further uses the protein export machinery; (3) genetic profiling suggested that tGFP-op and NES-tGFP-op overload the proteasome and protein export machinery (*Figure 5D*); (4) expression levels of tGFP and NES-tGFP, which cause growth defects are far lower than that of GFP (*Figure 6B*); and (5) GFP-op isolated specific mutants that were not isolated in tGFP-op and NES-tGFP-op. *Figure 7F* provides a schematic model summarizing this idea. Only harmless proteins like GFP can be produced up to the ultimate expression levels that cause the protein burden, which seems to be related to actin functions. Other proteins, localized or aggregative, can be produced at far lower levels than the level which causes the protein burden because their overproduction overloads localization or protein degradation resources which are more limited than the protein synthesis resource.

Finally, we propose a 'barrel model' to explain the relationship between the capacity of intracellular processes and the limits of expression (*Figure 7G*). In order for the cell to maintain its vital functions, resources within the cell are distributed to processes such as synthesis, folding, degradation, and transport. Each process has a fixed capacity depending on the amount of resources allocated to it (represented by the size of the barrel in *Figure 7G*). Each protein is synthesized, folded, transported, and degraded while using the resources in the cell (shown with arrows in *Figure 7G*). Overproduction of proteins processed by each process causes an overload of each resource, thus stalling the processing of other proteins produced by that process and creating cellular dysfunction. If a protein is processed by more than one process, its overexpression will first cause an overload of the process with the smallest capacity. Therefore, the level of overexpression of a protein that causes growth inhibition is likely to be determined by the process with the lowest capacity out of the processes by which the protein is processed. The capacity of synthesis, where all proteins are processed, should be the largest, and therefore proteins that undergo only the synthesis processing, that is

those that fold on their own, localize to the cytoplasm, and do not undergo rapid degradation, are considered to have the highest expression limits (Such proteins are referred as gratuitous proteins here). Previous studie has suggested that GFP and Gpm1 are such proteins, and they cause growth inhibition when expressed up to 15% of total proteins (*Kintaka et al., 2016*; *Eguchi et al., 2018*). We define this proliferation inhibition effect as the protein burden. Protein burden is therefore thought to be caused by perturbations to protein synthesis. Our analysis of genetic interactions, however, did not provide a clear link to perturbations to the synthetic process; our results suggest that protein burden causes an unexpected perturbation of actin function. The details of this mechanism remain unclear at present, and further studies are needed. On the other hand, processes in which only a fraction of proteins are processed have a smaller capacity than synthesis, and the expression limits of proteins processed by those processes must be lower than the limits of protein burden. In fact, the expression limits of tGFP, which undergoes degradation by aggregation and ubiquitination (probably by misfolding), and NES-tGFP, which is transported outside the nucleus, were much lower than those of GFP. Since their expression limits were about 4% of GFP (*Figure 6B*), it can be estimated that their expression limits are about 0.6% of the total proteins. This amount may reflect the capacities of the processes by which these proteins are processed.

In conclusion, our genetic profiling successfully investigated the consequences of overproduction: overload of protein synthesis, nuclear export, and the proteasome. Mutants isolated in this study will be useful resources for further investigations into the general consequences of protein overproduction, especially the overloading of cellular processes.

# Materials and methods

## Strains and plasmids used in this study

The vector plasmid (pTOW40836), GFP-op plasmid (pTOW40836-TDH3$_{pro}$-GFP), tGFP-op plasmid (pTOW40836-PYK1$_{pro}$-tGFP), and NES-tGFP-op plasmid (pTOW40836-PYK1$_{pro}$-NES-tGFP) have been described previously (*Kintaka et al., 2016*; *Eguchi et al., 2018*). Other than SGA, strains BY4741 (*MATa his3Δ1 leu2Δ0 met15Δ0 ura3Δ0*) and BY4743 (*MATa/α his3Δ1/his3Δ1 leu2Δ0/leu2Δ0 LYS2/lys2Δ0 met15Δ0/MET15 ura3Δ0/ura3Δ0*) were used as wild-type strains in the analysis. The deletion mutant collection and temperature-sensitive mutant collection have been described previously (*Costanzo et al., 2016*). Yeast culture and transformation were performed as previously described (*Amberg and Burke, 2005*). A synthetic complete (SC) medium without uracil (Ura) or leucine (Leu) was used for yeast culture.

## Query strains

Y7092 (*MATa can1Δ::STE2pr-his5 lyp1Δ ura3Δ0 leu2Δ0 his3Δ1 met15Δ0*) was used for the query train in the SGA. Y7092-E2-Crimson (*MATa can1Δ::TDH3pr-E2-Crimson STE2pr-his5 lyp1Δ ura3Δ0 leu2Δ0 his3Δ1 met15Δ0*) was used for the query strain in the SGA with the GFP fluorescent measurement experiment.

## Synthetic genetic array (SGA) and colony size analysis

SGA and colony size analysis were performed as previously described (*Baryshnikova et al., 2010*). An empty plasmid (pTOW40836), and plasmids for overproducing GFP (pTOW40836-TDH3$_{pro}$-GFP), tGFP (pTOW40836-PYK1$_{pro}$-tGFP), and NES-tGFP (pTOW40836-PYK1$_{pro}$-NES-tGFP) were introduced into the deletion and temperature-sensitive mutant collections using robots to manipulate libraries in 1536-colony high-density formats. A query strain harboring each of the overexproduction plasmids and each of the *MATa* mutant strains harboring a different genetic alteration were mated on YPD. Diploid cells were selected on plates containing both selection markers (YPD + G418 + clonNAT) found in the haploid parent strains. Sporulation was then induced by transferring cells to nitrogen starvation plates. Haploid cells containing all desired mutations were selected for by transferring cells to plates containing all selection markers (SC –His/Arg/Lys + canavanine + thialysine + G418 + cnonNAT) to select against remaining diploids. To analyze the growth of each deletion strain with the plasmids, all custom libraries were replicated to SC–LU plates and grown for three days at 30°C.

The fitness of each strain was assessed as normalized colony size on agar plates. Measurements of fitness and calculation of genetic interaction scores for each strain from colony images on agar plates were performed using SGA-tool (http://sgatools.ccbr.utoronto.ca) (*Wagih et al., 2013*). The colony size was quantified and normalized as shown in *Figure 1—figure supplement 2*. Then the genetic interaction (GI) scores were calculated using the formula. GI score $(\varepsilon) = W_{AB} - W_A \times W_B$, where $W_{AB}$ is overproduction-plasmid/mutant fitness, $W_A$ is Vecvtor control/mutant fitness, and $W_B$ is set to one as shown in *Figure 1—figure supplement 2*. The GI scores were filtered using the defined confidence threshold (GI score, $|\varepsilon| > 0.08$), and p-value that reflects both the local variability of replicate colonies (four colonies/strain) and the variability of the strain sharing the same query or array mutation ($p < 0.05$) (*Baryshnikova et al., 2010*). This filtered data set was used for all analyses.

For GFP-op_positive mutants, we further filtered the mutants as follows. Initial GFP-op_positive 146 genes (147 mutants) contained genes involved in the His and Lys synthetic pathways. His and Lys (Arg) are used as marker genes for the SGA, and deletion mutants of *HIS*, *LYS*, and *ARG* genes should not grow in the SGA analysis. In fact, the colony sizes of these mutants in the vector control experiment were very small and were considered to be the carryover. We thus further isolated positively-interacting mutants by setting a threshold on the colony size of greater than 0.39 in the vector control experiment, selected according to the largest colony size (*ARG1*) among the *HIS*, *LYS*, and *ARG* mutants, to avoid the identification of false-positive GIs.

## Colony size mesurement

Colony size was measured by using the image analysis software SGA tools (http://sgatools.ccbr.utoronto.ca/) to determine accurate pixel colony sizes. Average values and standard deviations were calculated from at least six replications. Y7092 was used as the wild-type host strain.

## Liquid growth measurement

Cellular growth was measured by monitoring OD595 every 30 min using a model 680 microplate reader (BioRad). The maximum growth rate was calculated as described previously (*Moriya et al., 2006*). Average values and standard deviations were calculated from biological triplicates. BY4741 was used as the wild type host strain.

## GFP fluorescent measurement by typhoon

Two colonies/strain from the SGA were picked up and replicated to SC–U plates, and grown for two days at 30°C. To detect the fluorescence of the colony, plates were scanned by laser (GFP at 488 nm and E2-Crimson at 532 nm) using Typhoon 9210 (Amersham Biosciences). The image data were analyzed using GenePix Pro Software (Molecular Devices). Each colony was segmented by a circle with the same diameter, the fluorescence per pixel was detected, and the medians of the fluorescence in the circle were calculated. To normalize the intensity by plate, all medians were divided by the plate average median for GFP and E2-Crimson. The ratios of GFP/RFP were calculated, and the averages of the two colonies were used.

## Clustering analysis

The GI scores of GFP, tGFP, and NES-tGFP were clustered into 15 clusters by the hierarchical clustering (average) method using R (https://www.r-project.org).

## Enrichment analysis

Enrichment analysis was performed using the gene list tool on the *Saccharomyces* genome database (yeastmine.yeastgenome.org/yeastmine/bag.do) (*Cherry et al., 2012*).

## Microscope observation

Log-phase cells were cultivated in SC–Ura medium. Cell images were obtained and processed using a DMI6000 B microscope and Leica Application Suite X software (Leica Microsystems). GFP fluorescence was observed using the GFP filter cube. Cellular DNA was stained with 100 μg/ml Hoechst 33342 (H3570, ThermoFisher) for 5 min and observed using the A filter cube. BY4741 was used as the host strain.

## Quantification of GFP expression level

The total protein was extracted from log-phase BY4741 cells harboring overproduction plasmids with NuPAGE LDS sample buffer (ThermoFisher NP0007) after 0.2 N NaOH treatment for 5 min (*Kushnirov, 2000*). For each analysis, the total protein extracted from 0.1 optical density unit of cells OD600 1.0 was used. The extracted protein was labeled with Ezlabel FluoroNeo (WSE-7010, ATTO), as described in the manufacturer's protocol, and separated by 4–12% SDS-PAGE. Proteins were detected and measured using a LAS-4000 image analyzer (GE Healthcare) in SYBR–green fluorescence detection mode, and Image Quant TL software (GE Healthcare). The intensity of the 45 kDa band corresponding to Pgk1 and Eno1/2 was used as the loading control. To detect GFP, the SDS-PAGE-separated proteins were transferred to a PVDF membrane (ThermoFisher). GFP was detected using an anti-GFP antibody (11814460001, Roche), a peroxidase-conjugated second antibody (414151F, Nichirei Biosciences), and a chemiluminescent reagent (34095, ThermoFisher). The chemiluminescent image was acquired with a LAS-4000 image analyzer in chemiluminescence detection mode (GE Healthcare). For the estimation of relative GFP levels, the intensities of corresponding GFP bands were normalized using the loading control described above.

## Cell fractionation and detection of ubiquitination

BY4743 cells with an overproduction plasmid were cultured overnight at 30°C in 25 ml SC–Leu/Ura medium. Cells were collected, cells were suspended in 1 mL of lysis buffer 10 mM Phosphate buffered saline (pH7.4), 0.001% Tween20, Halt Protease Inhibitor Cocktail (78425, ThermoFisher). Glass beads were added to the cell suspension and the tube was vortexed three times for 2 min. The sample was chilled on ice for 3 min between vortexing. The cell lysate was centrifuged at 20,000 × g for 10 min, and the supernatant was transferred to another tube. The precipitates were washed five times with 1 mL of PBST. The final precipitates were suspended in 100 µL of PBST. The sample was treated with NuPAGE sample buffer (NP0007, ThermoFisher) at 70°C for 10 min., and the proteins were separated by SDS-PAGE. Total protein and GFP were detected by Ezlabel FluoroNeo and western blotting with GFP antibodies as described above. Detection of ubiquitin was performed by western blotting same as the one of GFP except that the anti-ubiquitin antibody (P4D1, Santa Cruz) was used.

## Proteome analysis

Protein samples were precipitated by methanol/chloroform method and resolved in 0.1 M Tris-HCl (pH 8.5) containing 8 M urea. After reduction with DTT and alkylation with iodoacetoamide, urea concentrations were diluted by 4-fold with 0.1 M Tris-HCl (pH 8.5) and then digested into peptides by trypsin (Promega, Wisconsin, WI). Digested samples were centrifugated at 20,000 × g for 10 min and the tryptic peptides in supernatant were analysed by liquid chromatography-tandem mass spectrometry (LC-MS/MS) system consisting of a *DiNa* nano LC (KYA technologies, Tokyo, Japan) and a LTQ-Orbitrap XL mass spectrometer (Thermo Fisher Scientific, Waltham, MA). Acquired MS/Ms spectra were subjected to database search against protein sequences downloaded from the *Saccharomyces* Genome Database (http://www.yeastgenome.org/) by SEQUEST alogorithm. The number of peptide-spectrum match (PSM) for each protein, which fulfill the criteria of false discovery rate below 1%, was listed in *Figure 6—source data 1*.

## High-dimensional morphological analysis

Morphological data of cells cultured were acquired as previously described (*Ohya et al., 2005*). Briefly, logarithmic-phase BY4741 cells harboring plasmids grown in SC–Ura medium were fixed and were triply stained with FITC-ConA, rhodamine-phalloidin, and 4,6-diamidino-2-phenylindole to obtain fluorescent images of the cell-surface mannoprotein, actin cytoskeleton, and nuclear DNA, respectively. Images of at least 200 individual cells were acquired and processed using CalMorph (version 1.2). All of the statistical analyses were performed with R. To statistically test the morphological differences among four strains, we conducted one-way ANOVA of the generalized linear model (GLM) for each of 501 morphological parameters. Probability density functions (PDFs) and accompanying link functions in the GLM were assigned to each trait as described previously (*Yang et al., 2014*). Difference of the four strains (n = 5) was incorporated as the explanatory variable into the linear model. We assessed a dispersion model among the strains in the linear models

for the 501 parameters by Akaike information criterion (AIC) and set 110 models (parameters) as a different dispersion model because of lower AIC than that of a single dispersion model. Applying one-way ANOVA among the four strains to all 501 parameters, 51 of the 501 parameters were found to differ significantly at false discovery rate (FDR) = 0.01 by the likelihood ratio test (Likelihood ratio test in *Figure 7—source data 1*). Maximum likelihood estimation, likelihood ratio test, and the estimation of FDR were performed using the gamlss, lrtest, and qvalue functions in the gamlss (*Stasinopoulos and Rigby, 2007*), lmtest (*Zeileis and Hothorn, 2002*), and qvalue (*Storey, 2002*) R package. By Wald test at FDR = 0.01, 16, 17, and 24 of the 501 traits were detected to have a significant difference from wild-type in GFPy66g, Gpm1, and Gpm1-m, respectively (Q value of Wald test in *Figure 7—source data 1*). Of the 16 parameters detected in GFPy66g, 14 parameters were grouped into four independent morphological features by four principal components (explaining 60% of the variance) extracted from principal component analysis for the Z values of 109 replicates of *his3Δ* (*Suzuki et al., 2018*) as described previously (*Ohnuki et al., 2012*), and were used for the illustration of morphological features (*Figure 7E*, Morphological features in in *Figure 7—source data 1*).

## Acknowledgements

We thank members of the Moriya lab, the Boone lab and the Andrews lab for advice and helpful discussions. This work was partly supported by JSPS KAKENHI grant numbers 15KK0258, 17H03618, 18K19300, 20H03242.

## Additional information

### Funding

| Funder | Grant reference number | Author |
|---|---|---|
| Japan Society for the Promotion of Science | 17H03618 | Hisao Moriya |
| Japan Society for the Promotion of Science | 15KK0258 | Hisao Moriya |
| Japan Society for the Promotion of Science | 18K19300 | Hisao Moriya |
| Japan Society for the Promotion of Science | 20H03242 | Hisao Moriya |
| Japan Society for the Promotion of Science | 19H03205 | Yoshikazu Ohya |

The funders had no role in study design, data collection and interpretation, or the decision to submit the work for publication.

### Author contributions

Reiko Kintaka, Formal analysis, Investigation, Visualization, Writing - review and editing; Koji Makanae, Shotaro Namba, Keiji Kito, Investigation; Hisaaki Kato, Formal analysis, Investigation; Shinsuke Ohnuki, Formal analysis, Visualization, Writing - review and editing; Yoshikazu Ohya, Resources, Supervision, Methodology, Writing - review and editing; Brenda J Andrews, Resources, Supervision, Funding acquisition, Project administration; Charles Boone, Resources, Supervision, Funding acquisition, Project administration, Writing - review and editing; Hisao Moriya, Conceptualization, Supervision, Funding acquisition, Investigation, Visualization, Writing - original draft, Project administration, Writing - review and editing

### Author ORCIDs

Koji Makanae https://orcid.org/0000-0002-8560-4561
Keiji Kito https://orcid.org/0000-0002-9057-1688

Yoshikazu Ohya 🄳 https://orcid.org/0000-0003-0837-1239
Hisao Moriya 🄳 https://orcid.org/0000-0001-7638-3640

### Decision letter and Author response

Decision letter https://doi.org/10.7554/eLife.54080.sa1
Author response https://doi.org/10.7554/eLife.54080.sa2

## Additional files

### Supplementary files

• Supplementary file 1. Enrichment analysis of genes isolated in GFPop SGA analysis; associated with *Figure 2D*.

• Supplementary file 2. Enrichment analysis of genes isolated by GFP expression level; associated with *Figure 4—figure supplement 2A*; *Figure 4—figure supplement 2B*; *Figure 4C*; *Figure 4D*; *Figure 5—figure supplement 2*; *Figure 5E*; *Figure 6—figure supplement 1.A*.

• Supplementary file 3. Enrichment analysis of genes isolated as 14 mutants in *Figure 4—figure supplement 3A*.

• Supplementary file 4. Enrichment analysis of genes in each cluster in *Figure 5C and D*.

• Transparent reporting form

### Data availability

All data generated or analysed during this study are included in the manuscript and supporting files.

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
