## [Decision Letter]

**Acceptance summary:**

The study enhances our understanding of cellular effects of protein load.

**Decision letter after peer review:**

Thank you for submitting your article "Genetic Profiling of Protein Burden and Nuclear Export Overload" for consideration by *eLife*. Your article has been reviewed by three peer reviewers, and the evaluation has been overseen by a Reviewing Editor and Patricia Wittkopp as the Senior Editor. The reviewers have opted to remain anonymous.

The reviewers have discussed the reviews with one another and the Reviewing Editor has drafted this decision to help you prepare a revised submission.

Summary:

In "Genetic Profiling of Protein Burden and Nuclear Export Overload", Kintaka et al. performed a synthetic genetic array screen to explore the genetic interactions between mutants (DMA and TSA libraries) and protein burden. They used the 2 micron plasmid of *Saccharomyces cerevisiae* to explore the genetic interactions in response to three different protein burdens/overloads: (i) regular GFP; (ii) triple-GFP (tGFP), a much bigger protein that they found created some aggregates when highly expressed; and (iii) the nuclear export signal-containing tGFP (NES-tGFP). A similar study has already been published in this journal (Farkas et al., 2018), where the genetic interaction between the DMA library and yEVenus burden was explored. The present work, however, is much more extensive: it includes four biological and two experimental repeats, a second mutant library (TSA), different types of burden, and, most importantly, it checks the effect of higher burden levels using a stronger promoter.

Essential revisions:

Fundamental:

Overall, this work increases the knowledge about processes involved in protein stress alleviation. Yet, while the manuscript describes interesting phenomena, it does not provide follow-up findings to explain the main observations and therefore remains descriptive in nature. Here are concrete suggestions to this end.

1) In general, since there is no systematic follow-up on the main findings, it is difficult to appreciate how cells deal with the increased protein burden. In particular, as the authors noted, their findings are significantly different from other studies using similar reporter systems. Additional work is needed in order to substantiate the different pathways.

2) The authors overexpress GFP that does not have any physiological activity in yeast cells and therefore considered non-harmful. Yet, overexpression of tGFP shows genetic interaction with the proteasome. This may imply that tGFP undergoes post-transcriptional modification(s), such as ubiquitylation and this must be taken into consideration when attempting to explain the data. Accordingly, the authors need to examine the ubiquitylation status of their constructs, especially in proteasome where ubiquitin conjugates accumulate. Abolishing GFP ubiquitylation (using a lysineless variant, for example), may mitigate or abolish genetic interactions with proteasome subunits, suggesting a role for ubiquitin during overexpression.

3) tGFP-op levels are !~10 fold lower than GFP-op and yet only tGFP forms aggregates and affects cell viability in proteasome mutants. This is an interesting observation that requires some explanations. How tGFP-op is related to its tendency to aggregate and whether it is related to the proteasome activity should be verified before drawing a conclusion.

4) A main limitation of this work is the use of "tug of war" plasmid to generate the burden. This means that the plasmid copy number, and with it the GFP levels, will change across different mutations. The authors found, for example, that GFP levels are lower in many of the Mediator mutants. The reaction of the mutant cells to the burden should be measured not only by the growth effect but also by the burden level. The latter can be approximated by GPF levels or by the plasmid copy number. The authors report that in -Leu/Ura conditions, there are 30 copies of the plasmid. This statement is correct only for WT, and needs to be measured or estimated for the different mutations. This estimation, again, can be done by measuring the GFP levels of the different mutants. For the GFP set, they indeed used a method to measure the GFP levels. It would have helped to understand the mutants in Figure 3, if the GFP levels had been shown – this could easily be done in a supplementary figure. Fluorescence levels were, however, not done in the tGFP and NES-GFP experiments, or at least they are not being reported. It is important and probably critical to successful publication, that GFP levels be measured and reported, to help interpret the results correctly. At a minimum, this GFP levels should be given for those mutants above the GI threshold.

5) When the authors compared their results with those of Farkas et al., they didn't find any correlation. Neither study, incidentally, quantified GFP levels. Also, the present study used a plasmid with much higher expression levels, which will inevitably have obscured comparability. Farkas and co-workers used yEVenus and they showed that "yEVenus binds weakly, but significantly to certain molecular chaperones…". This binding, that may be unique to yEVenus, could also be part of the reason for the apparent differences in results. Farkas et al. showed that the burden effect was reduced by adding more AA to the media. Checking the AA effect on the GFP burden would have helped to reconcile whether the difference in results is a reflection of a different kind of burden, or whether it is mainly the results of a growth effect that wasn't normalized by GFP levels.

Presentation:

The writing of the manuscript and the interpretation of the data needs considerable expansion. The methods and the results are not described clearly. Please see major comments below.

1) Clarify the definition of genetic interactions.

1a) The term “genetic interactions” should be defined in the Introduction in a way that is specific to how it is used in this study, e.g. "In this study we screened for gene knockouts or knockdowns that had different impacts on growth depending on whether a green fluorescent protein was overexpressed."

1b) Clarify the meaning of positive and negative interactions, specifically whether the mutations are deleterious or beneficial. A positive interaction could either mean that GFPop alleviates a growth defect or enhances a growth advantage. Which one happens more? Probably the former but this needs to be clearer. This should be broken down in the figure, e.g. what fraction of orange dots signify a growth defect being alleviated vs. a growth advantage being enhanced? Can the authors use 4 colors, blue, light blue, orange, light orange? Since the goal here is to understand the biology of the cell and how GFPop changes it, these details seems important.

2) Clarify the protein burden.

2a) It is not a good idea to assume readers are familiar with previous publications using the tow system. Thus, most readers will assume that all mutant strains are expressing the same amount of GFP plus or minus some kind of noise (e.g. plasmid copy number variation due to unequal cell division). Please explain this in more detail in the Results section.

2b) Also, the term protein burden needs to be continuously explained. It is easy to interpret that term as meaning “the growth defect imposed by GFPop”. But it probably refers to the number of GFPop molecules produced. Is that right? Rather than using the term, “protein burden”, sometimes it would be OK to spell out what is meant, e.g. the number of GFPop molecules that burden the cell.

2c) An interesting question that arises is whether, in negative interactions, the expression of GFPop is enhancing the growth defect of the gene knockout/knockdown or the gene knockout is increasing the protein burden, e.g. increasing the level to which the tow system is able to express GFP. The authors have an impressive method of disentangling these two hypotheses, which they explain in Figure 4. But paragraph two of subsection “Investigation of GFP expression levels of mutants” of the paper, where these possibilities are enumerated, are unclear. One has to go back and work out many details including the possible types of positive and negative interactions, and how the tow system worked. A diagram or cartoon should be presented earlier in the paper, perhaps as Figure 1, which explains the logic showing the different possibilities that could be happening inside of cells and how the authors plan to disentangle them. The authors could depict 4 double-mutant cells, for example, one with a high level of GFP but a lower growth rate than either the GFPop strain or the mutant strain. Then they could explain in the legend the hypothesis as to what is happening inside this cell. Maybe a figure is unnecessary. But some more detailed explanation of how this system works and how the authors plan to us it to disentangle the possible things happening inside of cells should come up much earlier in the paper.

3) The growth regime is not clearly explained.

3a) The text in the first part of the Results section is very brief and relies heavily on Figure 1 to explain the experimental set up. Perhaps add a few more sentences to guide the reader. For example, the comment in paragraph two of subsection “Isolation of mutants that have genetic interactions with GFP-op” about “colonies” does not make sense. It should have already been stated that growth is measured in colonies and not in liquid culture. Otherwise the word “colonies” comes from nowhere.

3b) On that note, Figure 1C is confusing because these particular measurements were taken in liquid culture. Why are two different culturing methods used? Since 1C is meant as a control to show how the GFP affects growth when no knockout/knockdown is present, shouldn't this control be performed using the same method as the rest of the experiments?

4) Why were the particular GI threshold levels chosen? Why were the cutoffs chosen, including 0.08 in Figure 1 and 0.2, to define GFPop-positive and negative?

5) Why better correlation across conditions than within conditions? Figure 2A shows that even when taking only GI that exceed the 0.8 threshold, the correlation between replicates is less than 0.5 in the DMA -Ura condition. But Figure 2B shows that when comparing DMA -Ura to DMA -Ura/-Leu the correlation is greater than 0.5. How come the correlation between replicates is less than the correlation between these two different conditions? Is it because by averaging multiple replicates you get a better sense of the true GI value? This should be addressed.

6) Too many abbreviations. There are so many terms in this paper that are used to capture important concepts, e.g. positive interaction, protein burden, TMA, GFPunit_L, GFPunit_H, GFPop_positive, GFPop_negative. The reader loses track of how all of these things are related. The authors should not rely on these abbreviations so much and talk though these relationships, e.g. "Mutants with growth defects that were enhanced by GFP overexpression were also more likely to produce less GFP, indicating that the limit of GFP overproduction in these cells was lower than in other cells." Sentences like these would be so helpful in explaining what is actually going on.

7) Sometimes the Results section read a little bit like a list.

7a) This is a common issue in studies that report GO terms from different groups. Figure 3 is particularly list-like. The balance of the paper should shift away from listing GO categories and towards explaining and interpreting what is happening inside of cells as in paragraph two of subsection “Investigation of GFP expression levels of mutants”, and in Figure 6. For example, the authors were nicely able to do more with the actin and cell bud genes and show that indeed these cells had aberrant morphology. Also Figure 6F where the authors digest all of this information into a hypothesis about what is happening inside of cells was nice. Problematically, that hypothesis is unclear and will only become clear once points 1 and 2 above are addressed.

7b) Also, in 6F the relationship to the proteasome was not as clear as was the perturbation of actin. Could the proteasome relationship be explained more clearly?

Final thoughts: In sum, the authors' goal seems to be to go beyond listing GO categories to talk about how the protein burden affects cell biology. They need to rework the paper a bit in order to achieve this goal.

8) In the NES-tGFP experiment (and only in this experiment), they found strong interaction with the nucleus export machinery. This result isn't surprising per se but it's a good proof of concept, and it can be used as a control – showing that the system is working, which is worth mentioning. So this work could be an important resource to the field enabling a deeper understanding of protein burden origin. It should be emphasized.

[Editors' note: further revisions were suggested prior to acceptance, as described below.]

Thank you for submitting your article "Genetic Profiling of Protein Burden and Nuclear Export Overload" for consideration by *eLife*. Your article has been reviewed by three peer reviewers, and the evaluation has been overseen by a Reviewing Editor and Patricia Wittkopp as the Senior Editor. The reviewers have opted to remain anonymous.

The reviewers have discussed the reviews with one another and the Reviewing Editor has drafted this decision to help you prepare a revised submission.

As you can see, reviewer 2, raised outstanding issues regarding the mechanism. Originally they felt that more experiments are needed. However, in the discussion that followed they agreed to allow the authors to deal with them by adding statements that further studies to reveal the mechanisms are needed. It is up to you to decide whether you would like to add more experiments or language changes.

Here is a part of the discussion that elaborate on the proposed experiments, which could be useful to you:

"One model that I can think of, which is based on recent literature, is that tGFP undergoes ubiquitylation and then a portion of the ubiquitylated protein is sequestered into large aggregates. Proteasomes are been recruited to these aggregate but the kinetics of degradation is altered. This results in overall reduced proteasome capacity, shown through genetic interaction with proteasome subunits. I proposed to the authors to address the connection between overexpression and aggregation.

In my second revision, I suggested two experiments:

1) to IP tGFP from the pellet and show that indeed it is the protein that undergo ubiquitylation. I believe that technically this experiment is essential.

2) To reduce overall ubiquitylation (through the overexpression of mutant R48 ubiquitin) and test if tGFP is still aggregated and if this has an effect on cell growth.

One can think of alternative experiments, but the point is that the basis to the effects the authors observed upon overexpression of tGFP is still unclear to me."

Otherwise, please re-phrase some of the statements regarding the ubiquitylation of tGFP and the possible mechanism of tGFP function. Regarding the former, showing poly-Ub chains in the pellet does not prove that tGFP is indeed as the substrate. Regarding the latter, according to the accumulating data, the mechanism is unlikely to be a general effect of overexpression that leads to the proteasome. A statement about the effect of aggregates on proteasomal degradation could also help clarifying the issue. Possibly the depletion of essential factors like molecular chaperones?

Reviewer #1:

The authors have addressed all of my previous comments, moreover, this is an impressive revision. The authors add new figures and text that clarify their methods and previous findings. The authors also add new analyses, including biochemical analyses of some of their overexpression strains. Finally, the authors add a new figure and synthesize their results into a model which describes the different ways cells might be affected by overexpressed proteins.

The revised paper represents a massive amount of experiments and careful thought about how the cell responds to overexpressed proteins. The paper sheds new light on this question.

Reviewer #2:

The authors nicely addressed my first comment. however, in my opinion, issues 2 and 3 require further clarification:

In respond to my comments regarding the level of expression and PTM of tGFP, the authors tested whether tGFP aggregates were ubiquitinated and concluded that overexpressed tGFP but not GFP forms ubiquitinated aggregates in cells. They hypothesized that tGFP-op causes an overload of the proteasome because tGFP is frequently misfolded, ubiquitinated and degraded by the proteasome. This may be the cause of the negative genetic interactions between tGFP-op and the proteasome mutants.

I suggest that this hypothesis should be further clarified, since aggregation of tGFP is at the center of the manuscript and without having a mechanistic explanation to its function, the significance of some the authors findings is unclear. Generally, the overexpression of misfolded proteins (even large ones) per se does not inhibit the proteasome in yeast cells. Alternatively, the recruitment of proteasomes to protein aggregates may abrogate their function. Since the proteasome harbor several ubiquitin receptors, it is possible that protesomes interact with aggregated proteins through conjugated ubiquitin chains. This could be tested for tGFP by isolating it from aggregates by IP and looking for ubiquitylation. Furthermore, I accept that having lysineless GFP might not be the best approach to tackle the issue. Yet, the authors could test the effect of conditional overexpression of lys48 ubiquitin mutant on aggregate formation, proteasome function and/or cell viability in cells overexpressing GFP or tGFP. This type of experiments should shade some light on the molecular basis for tGFP aggregation and the effect on cell growth.

Reviewer #3:

The revise version of "Genetic Profiling of Protein Burden and NuclearExport Overload", is greatly improved. the logic is easier to follow, and there are more illustrations. The comparison between their and Farkas' results are dipper, and most important to my opinion, they now have the GFP measurements for all the conditions, and those measurements now better integrated into the text and figures.

Their final model about the different ways protein overproduction can affect growth rate is very nice. they display some of the interesting biological questions that rise up from their screening, include the interaction between actin and protein burden, and that over-expression tGFP lead to aggregate and proteasome stress. They also did some follow-ups experiments to both of them, but they didn't reach to a biological understanding about the origin of those interesting observations: way Actin is so important to protein overproduction? and the reason for tGFP aggregates. It would have been nice to get a better understanding to at list one of them but many times interesting questions are as important as answers, and it is very extensive and interesting screen. So, they answered most of the question, and in my opinion the current wark important and good enough for publication.

---

## [Author Response]

Essential revisions:Fundamental:Overall, this work increases the knowledge about processes involved in protein stress alleviation. Yet, while the manuscript describes interesting phenomena, it does not provide follow-up findings to explain the main observations and therefore remains descriptive in nature. Here are concrete suggestions to this end.1) In general, since there is no systematic follow-up on the main findings, it is difficult to appreciate how cells deal with the increased protein burden. In particular, as the authors noted, their findings are significantly different from other studies using similar reporter systems. Additional work is needed in order to substantiate the different pathways.

We believe that the lack of reproducibility between our experimental results and those of Farkas et al. reflects the difficulty in obtaining genetic interactions when querying weak overexpression – protein burden. The rationale for this is as follows.

We found that reproducible genetic interactions are unlikely to be obtained from moderate overexpression of EGFP, as shown in Figure 2A and Figure 2—figure supplement 1. Moderate overexpression in our experiments was performed by expressing yEGFP from the *TDH3* promoter cloned on a 2µ plasmid vector (Figure 2—figure supplement 3C). Similar to Farkas' conditions, overexpression of yEGFP from the *HSC82* promoter cloned on a 2µ vector (Figure 2—figure supplement 3D) elicits milder growth inhibition under the moderate overexpression conditions. This suggests that the yEVenus overexpression condition used by Farkas et al. is milder than our -Ura condition and does not result in reproducible genetic interactions (reproducibility was not mentioned in Farkas et al., 2018 because Farkas et al. did not perform multiple experimental trials.)

Another basis for this is that the distribution of genetic interaction scores obtained by Farkas et al. is very narrow (see Figure 2—figure supplement 3A, B), and the threshold of the genetic interaction score for the presence of genetic interaction is 0.05. Our threshold is 0.2. If this threshold were used in the Farkas et al. analysis, only about 10 genes would show interaction.

Furthermore, we have confirmed the genetic interaction of GFP-op with the negative genes obtained by Farkas et al. and our analysis in individual experiments (Figure 2—figure supplement 4). While none of the seven negative genes obtained by Farkas et al. showed a negative genetic interaction with GFP-op, six of the seven negative genes we obtained reproduced the negative interaction with GFP-op.

Thus, we believe that this discrepancy reflects the difficulty in obtaining genetic interactions when a weak overexpression – protein burden is used as a query.

On the other hand, we cannot rule out the possibility that this result arises from differences in the experimental system we used, especially between the fluorescent proteins EGFP and Venus. What we want to know in this study, however, is not the physiological state that occurs when individual proteins such as EGFP and Venus are overexpressed, but the general physiological state of protein burden caused by the extreme overexpression of non-toxic gratuitous proteins, using EGFP and Venus as models. Therefore, there is no point in investigating genetic interactions (or vice versa) that can only be observed with either an excess of Venus or an excess of EGFP. We do not feel it makes sense to pursue this further in the present study, as it only investigates the differences in the characteristics of the two proteins.

In this regard, we have performed a multidimensional morphological analysis in the overexpressing strains, as shown in Figure 7. We show that actin mislocalization is caused not only by an excess of EGFP but also by an excess of another protein, Gpm1, indicating that perturbations to actin are more commonly caused by protein burden.

2) The authors overexpress GFP that does not have any physiological activity in yeast cells and therefore considered non-harmful. Yet, overexpression of tGFP shows genetic interaction with the proteasome. This may imply that tGFP undergoes post-transcriptional modification(s), such as ubiquitylation and this must be taken into consideration when attempting to explain the data. Accordingly, the authors need to examine the ubiquitylation status of their constructs, especially in proteasome where ubiquitin conjugates accumulate. Abolishing GFP ubiquitylation (using a lysineless variant, for example), may mitigate or abolish genetic interactions with proteasome subunits, suggesting a role for ubiquitin during overexpression.

We initially predicted that tGFP was just three GFPs bound together and that the effects of overexpression would be no different from those of GFP.

However, our experiments revealed that tGFP-op induces a distinctly different physiological state than GFP-op, because the expression limit of tGFP is only a few percent of that of GFP (original Figure 5F, new Figure 6B), tGFP-op and GFP-op genetically interact with different sets of genes (Figure 5C,D), and only tGFP forms giant aggregates in the cell (original Figure 5E, new Figure 6C).

While these findings are interesting, we did not see much point in pursuing this further, as tGFP is a rather specific, artificial protein with "three GFPs in a row".

On the other hand, in response to the reviewer's comments, we thought that pursuing these findings might reveal the properties of proteins that exhibit toxicity in a different way than protein burden. In particular, tGFP may be considered a model for high-molecular-weight proteins (75 kDa) or for proteins with a repetitive structure.

The significance of ubiquitination may be tested by analysis of EGFP without lysine, as pointed out by the reviewer. However, EGFP contains 20 lysines and it has been reported that their replacement with arginine induced misfolding (PMID: 22792305). Therefore, it is not appropriate to use lysineless EGFP to investigate the significance of ubiquitination, as overexpression of lysineless EGFP may cause unexpected results.

Instead, we used biochemical methods to test whether tGFP aggregates were ubiquitinated: GFP-op and tGFP-op cells were disrupted, separated into soluble and insoluble fractions, and then western blotted with anti-ubiquitin antibodies, respectively. The results showed a highly ubiquitinated high molecular weight ladder in the insoluble fraction of the tGFP-op strain (new Figure 6D). Western blotting with anti-GFP antibodies showed a high molecular weight ladder in the insoluble fraction of GFP-op as well as tGFP-op (new Figure 6D). However, the high-molecular-weight ladder found in the insoluble fraction of GFP-op was not detected by the anti-ubiquitin antibody. These results suggest that overexpressed tGFP but not GFP forms ubiquitinated aggregates in cells.

Our hypothesis is that tGFP-op causes an overload of the proteasome because tGFP is frequently misfolded, ubiquitinated and degraded by the proteasome. This may be the cause of the negative genetic interactions between tGFP-op and the proteasome mutants.

On the other hand, based on the detection of genetic interactions and ubiquitination, overexpression of single GFP did not cause ubiquitination or load on the proteasome. This may be related to the fact that tGFP is larger than GFP and has features such as a repeating structure of the same molecule strung together.

The above results have been added to new Figure 6D, and we created a new section on the analysis of tGFP.

3) tGFP-op levels are !~10 fold lower than GFP-op and yet only tGFP forms aggregates and affects cell viability in proteasome mutants. This is an interesting observation that requires some explanations. How tGFP-op is related to its tendency to aggregate and whether it is related to the proteasome activity should be verified before drawing a conclusion.

As described above, we performed a biochemical analysis of tGFP. We detected a high molecular weight ladder that was ubiquitinated in the insoluble fraction only in tGFP-op. This ladder is likely to be ubiquitinated tGFP because it is at a higher molecular weight than tGFP.

Although it is currently only a speculation, we believe that this difference is due to the higher molecular weight of tGFP compared to GFP and the presence of a repeating structure within the molecule. A larger molecule may increase the likelihood of misfolding during translation. The presence of a repeating structure may increase intermolecular interactions and trigger the creation of large aggregates. These abnormal proteins may be processed by degradation by the ubiquitin-proteasome system, thus creating a load on these pathways.

While we did write a short description of the expected nature of tGFP in the original manuscript in the Discussion, we have added a further statement above.

4) A main limitation of this work is the use of "tug of war" plasmid to generate the burden. This means that the plasmid copy number, and with it the GFP levels, will change across different mutations. The authors found, for example, that GFP levels are lower in many of the Mediator mutants. The reaction of the mutant cells to the burden should be measured not only by the growth effect but also by the burden level. The latter can be approximated by GPF levels or by the plasmid copy number. The authors report that in -Leu/Ura conditions, there are 30 copies of the plasmid. This statement is correct only for WT, and needs to be measured or estimated for the different mutations. This estimation, again, can be done by measuring the GFP levels of the different mutants. For the GFP set, they indeed used a method to measure the GFP levels. It would have helped to understand the mutants in Figure 3, if the GFP levels had been shown – this could easily be done in a supplementary figure. Fluorescence levels were, however, not done in the tGFP and NES-GFP experiments, or at least they are not being reported. It is important and probably critical to successful publication, that GFP levels be measured and reported, to help interpret the results correctly. At a minimum, this GFP levels should be given for those mutants above the GI threshold.

While we agree that there are limitations to overexpression using the gTOW plasmid, these limitations do not only exist for overexpression using the gTOW plasmid, but are potentially present in any overexpression experiment.

In the case of overexpression using 2µ plasmids, not only gTOW plasmids, the effect of mutants on plasmid copy number is inevitable. In fact, as shown in Figure 2—figure supplement 5, retention of the multicopy plasmid itself can negatively affect growth in replication mutants. A similar hidden bias may be present in the experiments of Farkas et al. that preceded the present study, as they used 2µ plasmids.

On the other hand, limiting is also present in promoter substitution, which is a possible alternative to overexpression using multicopy plasmids. The most commonly used overexpression system in yeast is the GAL1 promoter, but overexpression of EGFP by the GAL1 promoter from a single copy cannot cause growth inhibition, i.e., it cannot be overexpressed to the point of causing protein burden. Furthermore, even if SGA analysis is performed using overexpression by GAL1 promoter substitution as a query, it is not possible to avoid the effects of mutants that alter the expression levels from the GAL1 promoter. For example, positive interactions could be falsely detected in mutants of the basic transcription factor, such as those obtained in this study, because of the reduced degree of overexpression.

In other words, no matter what overexpression system is used, the risk that mutations affecting the system will be obtained as pseudo-positive or negative interactions cannot be ruled out.

As noted by the reviewer, we used GFP expression levels to address these issues (as discussed below, this principle is explained in the new Figure 4—figure supplement 1). In response to the reviewer's comments, we have added the GFP expression data to Figure 3. We also analyzed the expression levels of tGFP-op and NES-tGFP as well as GFP-op. These results supported that the nuclear transport mutants are sensitive to NES-tGFP overexpression. (New Figure 5E). It was also predicted that either the expression of tGFP is elevated in proteasome mutants or that those mutants are sensitive to tGFP overexpression (Figure 6A).

We have added these results as a new figure (Figure 5E, Figure 5—figure supplement 2, Figure 6A) and added the relevant description to Results.

5) When the authors compared their results with those of Farkas et al., they didn't find any correlation. Neither study, incidentally, quantified GFP levels. Also, the present study used a plasmid with much higher expression levels, which will inevitably have obscured comparability. Farkas and co-workers used yEVenus and they showed that "yEVenus binds weakly, but significantly to certain molecular chaperones…". This binding, that may be unique to yEVenus, could also be part of the reason for the apparent differences in results. Farkas et al. showed that the burden effect was reduced by adding more AA to the media. Checking the AA effect on the GFP burden would have helped to reconcile whether the difference in results is a reflection of a different kind of burden, or whether it is mainly the results of a growth effect that wasn't normalized by GFP levels.

As shown in Figure 2—figure supplement 3, we and Farkas both used a 2µ plasmid. Our gTOW plasmid showed a higher copy number only under the -Ura/Leu condition, and we believe that under the -Ura condition the copy number is similar to that of the normal 2µ plasmid and therefore the same as that of the YEplac181 plasmid used by Farkas and co-workers. Since Farkas et al. use a promoter for HSC82, it is likely to be weaker than expression from our *TDH3* promoter, but the degree of growth inhibition is not significantly different (Figure 2—figure supplement 3E). In other words, we believe that EGFP and Venus are expressed under our low-copy conditions at levels similar to the overexpression conditions of Farkas et al.

And, as shown in Figure 2—figure supplement 3, our experimental results are not consistent with those of Farkas et al. under these conditions as well. However, we believe that this lack of reproducibility reflects the difficulty in reproducibly obtaining genetic interactions under conditions of weak overexpression (Figure 2—figure supplement 1), as we already noted in section 1.

We adopted the high-copy condition for the present study because of its high reproducibility. The statistical analysis of the confident gene set showing genetic interactions under high-copy conditions did not suggest that GFP-op adversely affects HSP70-related proteostasis. Therefore, perturbations to HSP70-associated proteostasis should not be considered a general consequence of protein burden, or at least not a major consequence, and we do not see the significance of following up previous studies that follow that assumption in the present study.

The protein burden, as we see it, is a phenomenon caused by a "general" excess of a protein, not by a specific function of the overexpressed protein. If yEVenus binds to a specific chaperone and so depletes it, but EGFP does not because of its lack of binding to that chaperone, then we are not looking at a general phenomenon of protein burden. In fact, we are currently working on a study to evaluate the differences in cytotoxicity of EGFP, Venus, and other fluorescent proteins and will submit a paper soon, but this is beyond the scope of this paper and will not be mentioned in this paper.

On the other hand, since the burden effect on amino acid starvation can be immediately verified in our experimental setup, we performed the experiments. We measured the cost of GFP-op and yEVenus-op at diluted amino acid concentrations, as required by the reviewers (Figure 2—figure supplement 3G).

The results of Farkas et al. that the cost of yEVenus overexpression increased with decreasing amino acid concentration of the medium were confirmed in our experimental system. The cost of GFP overexpression also increased with decreasing amino acid concentration of the medium, but this increase was rather more pronounced than the increase in the cost of yEVenus. These results suggest that overexpression of GFP may impose a similar burden to that of yEVenus overexpression. On the other hand, the results also suggest that the effects of the excess of the two fluorescent proteins on cells are not entirely the same.

These results have been added to Figure 2—figure supplement 3G and the relevant description has been added to the Results.

Presentation:The writing of the manuscript and the interpretation of the data needs considerable expansion. The methods and the results are not described clearly. Please see major comments below.1) Clarify the definition of genetic interactions.1a) The term “genetic interactions” should be defined in the Introduction in a way that is specific to how it is used in this study, e.g. "In this study we screened for gene knockouts or knockdowns that had different impacts on growth depending on whether a green fluorescent protein was overexpressed."

Following the reviewer's advice, We have added the following text to the Introduction.

“To understand the physiological conditions caused by protein burden, we conducted a systematic survey of mutants that exacerbate or alleviate the growth inhibition caused by GFP overexpression. We surveyed genetic interactions between mutant strains and high levels of GFP overproduction (GFP-op) to genetically profile cells exhibiting this phenomenon. Here, if a mutation exacerbates growth inhibition by GFP-op, or if GFP-op exacerbates growth inhibition by the mutation, the mutation has a negative genetic interaction with GFP-op. Also, if a mutation alleviates growth inhibition caused by GFP-op, the mutation has a positive genetic interaction with GFP-op. If GFP-op relaxes the growth inhibition caused by the mutation, it is also detected as a positive genetic interaction.”

1b) Clarify the meaning of positive and negative interactions, specifically whether the mutations are deleterious or beneficial. A positive interaction could either mean that GFPop alleviates a growth defect or enhances a growth advantage. Which one happens more? Probably the former but this needs to be clearer. This should be broken down in the figure, e.g. what fraction of orange dots signify a growth defect being alleviated vs. a growth advantage being enhanced? Can the authors use 4 colors, blue, light blue, orange, light orange? Since the goal here is to understand the biology of the cell and how GFPop changes it, these details seems important.

The interpretation of the genetic interaction was added in the Introduction section as described above.

We did not anticipate this reviewer's perspective on positive genetic interactions. We appreciate your point of view.

We assumed that the background principle behind the "positive genetic interaction" would be that mutations mitigate the growth-inhibiting effects of GFP-op (Case 1). For example, as our analysis suggested, mutants of the basic transcription factor appear to lower the expression of GFP, thus lowering the growth inhibitory effect of GFP-op. Also, as we recently published in our paper, mutations in chromatin remodeling factors can alleviate the growth inhibition by GFP-op by lowering the transcription of a specific group of genes to create more resource space (Saeki et al., 2020).

On the other hand, it is certainly possible that GFP overexpression could alleviate the growth inhibition caused by the mutation (Case 2). Furthermore, if GFP-op further enhances the growth of strains whose growth is improved by the mutation (Case 3), this would be detected as a positive genetic interaction. With regard to Case 1 and 2, however, we cannot distinguish between them, as the present analysis only provides a value of "GFP-op and the mutation's growth-inhibiting effect is less than expected". With respect to Case 3, we can observe this as a phenomenon where "the growth of the GFP-op im the mutant is higher than the growth of the wild type with the vector".

However, as we will discuss later, our screening does not allow us to directly compare fitness between plates, as we correct for fitness within plates. Since GFP-op and vector controls are analyzed on separate plates, colony sizes on each plate cannot be directly compared. On the other hand, it would be very interesting to know if there are mutants in GFP-op that have a higher growth rate than the wild type (+vector control). Therefore, we investigated the mutant strains that showed a positive genetic interaction with GFP-op and a higher fitness in the vector control than in the wild type. Such mutants can be selected computationally by taking into account the inhibition of growth by GFP-op. By this revision, we assessed the fitness reduction by GFP-op, tGFP-op and NES-tGFP-op on the same plate (revised Figure 5B). The fitness reduction by GFP-op obtained there (0.78) was used to correct for colony size on GFP-op. The results of such a screening are shown in new Figure 4—figure supplement 3. As a result, 18 mutants were obtained. In addition, 14 mutants were obtained when the strains with higher than average GFPunit were screened. These were subjected to enrichment analysis and three components of the microtubule-binding protein DAM/DASH complex were found to be contained in these 14 mutants. At present, the molecular mechanism for why these mutants behave in this way is not clear. Individual analyses of these strains will be necessary in the future. This data is shown in new Figure 4—figure supplement 3 and 4D, and the relevant description has been added to Results.

In the process of this analysis, we found some errors in the graph in Figure 2D, which we have replaced. Some positive genes that were excluded from the analysis had been shown as orange circles in the original Figure 2D.

2) Clarify the protein burden.2a) It is not a good idea to assume readers are familiar with previous publications using the tow system. Thus, most readers will assume that all mutant strains are expressing the same amount of GFP plus or minus some kind of noise (e.g. plasmid copy number variation due to unequal cell division). Please explain this in more detail in the Results section.

In response to the reviewer's comments, we have added a detailed description of the gTOW system to new Figure 1—figure supplement 1. We have also added a detailed description of the gTOW system to Figure 4—figure supplement 1 to show how copy number variation by gTOW affects GFP expression levels.

2b) Also, the term protein burden needs to be continuously explained. It is easy to interpret that term as meaning “the growth defect imposed by GFPop”. But it probably refers to the number of GFPop molecules produced. Is that right? Rather than using the term, “protein burden”, sometimes it would be OK to spell out what is meant, e.g. the number of GFPop molecules that burden the cell.

Thanks for the advice. We define protein burden as the phenomenon of growth defects that occurs when a gratuitous protein is extremely overexpressed in the cell. Indeed, there was some ambiguity in the use of protein burden in the Results. We have changed the wording in this section to clarify that the protein burden is a growth inhibition phenomenon.

2c) An interesting question that arises is whether, in negative interactions, the expression of GFPop is enhancing the growth defect of the gene knockout/knockdown or the gene knockout is increasing the protein burden, e.g. increasing the level to which the tow system is able to express GFP. The authors have an impressive method of disentangling these two hypotheses, which they explain in Figure 4. But paragraph two of subsection “Investigation of GFP expression levels of mutants” of the paper, where these possibilities are enumerated, are unclear. One has to go back and work out many details including the possible types of positive and negative interactions, and how the tow system worked. A diagram or cartoon should be presented earlier in the paper, perhaps as Figure 1, which explains the logic showing the different possibilities that could be happening inside of cells and how the authors plan to disentangle them. The authors could depict 4 double-mutant cells, for example, one with a high level of GFP but a lower growth rate than either the GFPop strain or the mutant strain. Then they could explain in the legend the hypothesis as to what is happening inside this cell. Maybe a figure is unnecessary. But some more detailed explanation of how this system works and how the authors plan to us it to disentangle the possible things happening inside of cells should come up much earlier in the paper.

We agree that the original manuscript did not explain this part well enough and it was difficult for the reader to understand it. Therefore, we have added an interpretation of this experiment as new Figure 4B. We have also created a new figure (Figure 4—figure supplement 1) to illustrate the detailed background principles.

As described above, we also analyzed the GFP fluorescence for tGFP and NES-tGFP. This increased the number of concrete examples of gene sets belonging to the four categories, and we hope that the readers can now get a clearer picture of what these four categories mean.

The reviewer suggests that this analysis of GFP fluorescence and its interpretation be explained earlier in the Results. However, as the reviewer thinks, this analysis is a bit complicated and not easy to understand, so we would like to maintain the current order of explanation, first acquiring mutant strains by genetic interactions and then classifying them by GFP expression levels, in order to allow the readers to trace our thoughts.

3) The growth regime is not clearly explained.3a) The text in the first part of the Results section is very brief and relies heavily on Figure 1 to explain the experimental set up. Perhaps add a few more sentences to guide the reader. For example, the comment in paragraph two of subsection “Isolation of mutants that have genetic interactions with GFP-op” about “colonies” does not make sense. It should have already been stated that growth is measured in colonies and not in liquid culture. Otherwise the word “colonies” comes from nowhere.

We apologize for our lack of explanation. Following the reviewer's suggestion, we have added a text explaining how we measured growth (fitness) to the Results.

There is an established pipeline for calculating genetic interactions using the synthetic genetic array (e.g. Costanzo et al., 2016) and we have calculated genetic interactions accordingly.

On the other hand, because this analysis differs in some respects from the usual genetic interaction analysis, we have created a new figure (Figure 1—figure supplement 2) to explain the calculation of genetic interactions in this study. In the process of preparing this figure and explanation, we found some inaccuracies in the methods of the original manuscript, so we corrected them.

3b) On that note, Figure 1C is confusing because these particular measurements were taken in liquid culture. Why are two different culturing methods used? Since 1C is meant as a control to show how the GFP affects growth when no knockout/knockdown is present, shouldn't this control be performed using the same method as the rest of the experiments?

I agree with the reviewer's comment. We have measured growth at colony size and added it to a new figure (Figure 1C, D). We also performed colony-sized growth tests for tGFP-op and NES-tGFP-op and replaced Figure 5B with a new one, and changed the description according to the data.

4) Why were the particular GI threshold levels chosen? Why were the cutoffs chosen, including 0.08 in Figure 1 and 0.2, to define GFPop-positive and negative?

We chose these thresholds based on the previous publications, and to isolate confident (and strong) genetic interactions. We added the rationale to Results.

5) Why better correlation across conditions than within conditions? Figure 2A shows that even when taking only GI that exceed the 0.8 threshold, the correlation between replicates is less than 0.5 in the DMA -Ura condition. But Figure 2B shows that when comparing DMA -Ura to DMA -Ura/-Leu the correlation is greater than 0.5. How come the correlation between replicates is less than the correlation between these two different conditions? Is it because by averaging multiple replicates you get a better sense of the true GI value? This should be addressed.

We were simply showing this data to show that our mutant isolation method is effective.

Since the reproducibility between replicates at the 0.08 threshold for TSA is 0.70 (Figure 2A, TSA-0.08) and between conditions is 058 (Figure 2C), the reproducibility between replicates is higher in the case of TSA. On the other hand, indeed, for the DMA, the correlation is higher between conditions (0.70) than between replicates (0.62).

The reason for this is unclear, but as the reviewer points out, it may be an effect of averaging. We have therefore added the following statement to our Results:

“We note that there is a higher correlation between conditions at -Ura and -Leu/Ura than between replicates in DMA (Figure 2A and Figure 2B). The cause of this is unclear, but it may indicate that averaging between replicates yields values closer to the true GI score.”

6) Too many abbreviations. There are so many terms in this paper that are used to capture important concepts, e.g. positive interaction, protein burden, TMA, GFPunit_L, GFPunit_H, GFPop_positive, GFPop_negative. The reader loses track of how all of these things are related. The authors should not rely on these abbreviations so much and talk though these relationships, e.g. "Mutants with growth defects that were enhanced by GFP overexpression were also more likely to produce less GFP, indicating that the limit of GFP overproduction in these cells was lower than in other cells." Sentences like these would be so helpful in explaining what is actually going on.

Perhaps the abbreviation of GFPunit and its concept are the most confusing. We have added a new table of this as Figure 4B, as described above. We hope to avoid some confusion by referring the reader to this.

For the DMA and TSA in the figure, we have rewritten them to spell them out as much as possible.

7) Sometimes the Results section read a little bit like a list.7a) This is a common issue in studies that report GO terms from different groups. Figure 3 is particularly list-like. The balance of the paper should shift away from listing GO categories and towards explaining and interpreting what is happening inside of cells as in paragraph two of subsection “Investigation of GFP expression levels of mutants”, and in Figure 6. For example, the authors were nicely able to do more with the actin and cell bud genes and show that indeed these cells had aberrant morphology. Also Figure 6F where the authors digest all of this information into a hypothesis about what is happening inside of cells was nice. Problematically, that hypothesis is unclear and will only become clear once points 1 and 2 above are addressed.

The first half of this study is an attempt to identify the biological function of the genes we have isolated, using only statistical and objective indicators. We believe that enrichment analysis using GO (and publications) is currently the only way to objectively demonstrate the biological significance of a data set. The results of enrichment analysis must currently be a list. We feel, too, that there is certainly a need for a method of presenting the results of GO analysis in a graphical way that appeals to people's intuition.

On the other hand, we believe that Figure 3 is more than just a list of enrichment analysis results. Figure 3 allows the readers to know what exactly the names of the genes in each category of GO are. It will give more information than just the categories, especially to the experts. We also hope that by looking at this graph, the reader can see that there are multiple alleles in the temperature-sensitive strains, and that these alleles uniformly tend to show a positive/negative genetic interaction. Furthermore, we believe that the reader will understand that the gene cluster as a whole tends to exhibit positive/negative interactions beyond the noise of the experimental system. In addition to that, we have also added GFP expression levels to this graph in this revision. This graph should give the readers a clearer picture of the analysis performed in this experiment.

7b) Also, in 6F the relationship to the proteasome was not as clear as was the perturbation of actin. Could the proteasome relationship be explained more clearly?Final thoughts: In sum, the authors' goal seems to be to go beyond listing GO categories to talk about how the protein burden affects cell biology. They need to rework the paper a bit in order to achieve this goal.

That's right. We are only using enrichment analysis to objectively arrive at biological information from our data set.

In this revision, we analyzed GFP expression levels for tGFP and NES-tGFP, and we also performed biochemical analysis of tGFP (and GFP) overexpression strains. We hope that by presenting these analyses, we have clarified our original goal of understanding the physiological state of cells caused by overexpression by analyzing genetic interactions.

We have added this result as new Figure 5 and Figure 6 and have added relevant descriptions to the results.

We have further added a model as Figure 7G that clarifies the conceptualization of this study and a related description to Discussion.

8) In the NES-tGFP experiment (and only in this experiment), they found strong interaction with the nucleus export machinery. This result isn't surprising per se but it's a good proof of concept, and it can be used as a control – showing that the system is working, which is worth mentioning. So this work could be an important resource to the field enabling a deeper understanding of protein burden origin. It should be emphasized.

Thank you for the suggestion. Indeed, we also think this is the clearest result of the proof of concept. We have now also analyzed the data for GFP expression in NES-tGFP to show the results more clearly as Figure 5E. We have also rewritten the text to strengthen our argument.

[Editors' note: further revisions were suggested prior to acceptance, as described below.]

As you can see, reviewer 2, raised outstanding issues regarding the mechanism. Originally they felt that more experiments are needed. However, in the discussion that followed they agreed to allow the authors to deal with them by adding statements that further studies to reveal the mechanisms are needed. It is up to you to decide whether you would like to add more experiments or language changes.Here is a part of the discussion that elaborate on the proposed experiments, which could be useful to you:"One model that I can think of, which is based on recent literature, is that tGFP undergoes ubiquitylation and then a portion of the ubiquitylated protein is sequestered into large aggregates. Proteasomes are been recruited to these aggregate but the kinetics of degradation is altered. This results in overall reduced proteasome capacity, shown through genetic interaction with proteasome subunits. I proposed to the authors to address the connection between overexpression and aggregation.In my second revision, I suggested two experiments:1) to IP tGFP from the pellet and show that indeed it is the protein that undergo ubiquitylation. I believe that technically this experiment is essential.2) To reduce overall ubiquitylation (through the overexpression of mutant R48 ubiquitin) and test if tGFP is still aggregated and if this has an effect on cell growth.One can think of alternative experiments, but the point is that the basis to the effects the authors observed upon overexpression of tGFP is still unclear to me."Otherwise, please re-phrase some of the statements regarding the ubiquitylation of tGFP and the possible mechanism of tGFP function. Regarding the former, showing poly-Ub chains in the pellet does not prove that tGFP is indeed as the substrate. Regarding the latter, according to the accumulating data, the mechanism is unlikely to be a general effect of overexpression that leads to the proteasome. A statement about the effect of aggregates on proteasomal degradation could also help clarifying the issue. Possibly the depletion of essential factors like molecular chaperones?

We have made two revisions to answer this comment. We performed one experiment to identify proteins in the precipitated fraction of cells of tGFP-op and added a discussion of some mechanisms that might explain the negative genetic interaction between tGFP-op and the proteasome mutant predicted by the experimental results. We believe that this has allowed us to propose a clearer hypothesis about the mechanism of the negative genetic interaction between tGFP-op and the proteasome mutants.

Instead of performing the immunoprecipitation experiments suggested by the reviewer, we decided to perform a more direct experiment, i.e., systematic identification by LC-MS/MS of proteins in a precipitation fraction containing a large amount of aggregates generated by tGFP-op. We have added the results to the new Figure 6E and relevant description in Results. Instead, due to space constraints, we moved part of the older Figure 6D to Figure 6—figure supplement 1C.

As a result, proteasomes have been detected in the precipitation. Their amount was higher in tGFP-op than in the vector control, but it was still very small, so it was difficult to conclude that they were trapped in the aggregates and their intracellular concentration was reduced. On the other hand, Ssa1/Ssa2 chaperones, which are Hsp70, were detected in large quantities in the precipitation fraction of tGFP-op, as the reviewer had expected. It has been reported that Ssa1/Ssa2 is required for proteasome assembly, and we therefore believe that this result provides a promising model to explain the negative genetic interaction between tGFP-op and the proteasome mutants.

However, we believe that further experiments are needed to draw conclusions about the above as well. Therefore, as another revise, we have added our model for explaining the genetic interaction between tGFP-op and the proteasome mutant based on the experimental results as a new Figure 7—figure supplement 1, and added this explanation to the Discussion.

Reviewer #2:The authors nicely addressed my first comment. however, in my opinion, issues 2 and 3 require further clarification:In respond to my comments regarding the level of expression and PTM of tGFP, the authors tested whether tGFP aggregates were ubiquitinated and concluded that overexpressed tGFP but not GFP forms ubiquitinated aggregates in cells. They hypothesized that tGFP-op causes an overload of the proteasome because tGFP is frequently misfolded, ubiquitinated and degraded by the proteasome. This may be the cause of the negative genetic interactions between tGFP-op and the proteasome mutants.I suggest that this hypothesis should be further clarified, since aggregation of tGFP is at the center of the manuscript and without having a mechanistic explanation to its function, the significance of some the authors findings is unclear. Generally, the overexpression of misfolded proteins (even large ones) per se does not inhibit the proteasome in yeast cells. Alternatively, the recruitment of proteasomes to protein aggregates may abrogate their function. Since the proteasome harbor several ubiquitin receptors, it is possible that protesomes interact with aggregated proteins through conjugated ubiquitin chains. This could be tested for tGFP by isolating it from aggregates by IP and looking for ubiquitylation. Furthermore, I accept that having lysineless GFP might not be the best approach to tackle the issue. Yet, the authors could test the effect of conditional overexpression of lys48 ubiquitin mutant on aggregate formation, proteasome function and/or cell viability in cells overexpressing GFP or tGFP. This type of experiments should shade some light on the molecular basis for tGFP aggregation and the effect on cell growth.

To answer the concern of reviewer #2, we have performed two revisions as described above.